# A Region-Shrinking-Based Acceleration for Classification-Based Derivative-Free Optimization

## Abstract

Derivative-free optimization algorithms play an important role in scientific and engineering design optimization problems, especially when derivative information is not accessible. In this paper, we study the framework of classification-based derivative-free optimization algorithms. By introducing a concept called hypothesis-target shattering rate, we revisit the computational complexity upper bound of this type of algorithms. Inspired by the revisited upper bound, we propose an algorithm named "RACE-CARS", which adds a random region-shrinking step compared with "SRACOS" (Hu et al., 2017). We further establish a theorem showing the acceleration of region-shrinking. Experiments on the synthetic functions as well as black-box tuning for language-model-as-a-service demonstrate empirically the efficiency of "RACE-CARS". An ablation experiment on the introduced hyperparameters is also conducted, revealing the mechanism of "RACE-CARS" and putting forward an empirical hyperparameter-tuning guidance.

## 1 Introduction

In recent years, there has been a growing interest in the field of derivative-free optimization (DFO) algorithms, also known as zeroth-order optimization. These algorithms aim to optimize objective functions without relying on explicit gradient information, making them suitable for scenarios where obtaining derivatives is either infeasible or computationally expensive (Conn et al., 2009; Larson et al., 2019). For example, DFO techniques can be applied to hyperparameter tuning, which involves optimizing complex objective functions with unavailable first-order information (Falkner et al., 2018; Akiba et al., 2019; Yang & Shami, 2020). Moreover, DFO algorithms find applications in engineering design optimization, where the objective functions are computationally expensive to evaluate and lack explicit derivatives (Ray & Saini, 2001; Liao, 2010; Akay & Karaboga, 2012).

Classical DFO methods such as Nelder-Mead method (Nelder & Mead, 1965) or directional direct-search (DDS) method (Céa, 1971; Yu, 1979) are originally designed for convex problems. Consequently, their performance is compromised when the objective is nonconvex. One kind of well-known DFO algorithms for nonconvex problems is evolutionary algorithms (Bäck & Schwefel, 1993; Fortin et al., 2012; Hansen, 2016; Opara & Arabas, 2019). These algorithms have been successfully applied to solve optimization problems with black-box objective functions. However, theoretical studies on these algorithms are rare. As a consequence their performance lacks theoretical supports and explanations, making it confusing to select hyperparameters. In recent years, Bayesian optimization (BO) methods have gained significant attention due to their ability to efficiently optimize complex and expensive-to-evaluate functions (Snoek et al., 2012; Shahriari et al., 2015; Frazier, 2018). By leveraging probabilistic models, BO algorithms can guide the search process automatically, balancing the processes of exploration and exploitation. However, BO suffers from scalability issues when dealing with high-dimensional problems (Bickel & Levina, 2004; Hall et al., 2005; Fan & Fan, 2008). Another class of surrogate modelling algorithms called gradient approximation have also been extensively explored in the context of DFO (Nesterov & Spokoiny, 2017; Chen et al., 2019; Ge et al., 2022; Ragonneau & Zhang, 2023). These methods aim to estimate the gradients of the objective function using finite-difference or surrogate models combined with trust region. However, although some recent researches have shown the view that DFO algorithms using finite-difference are sensitive to noise should be re-examined (Shi et al., 2021; Scheinberg, 2022),

this kind of algorithms are intrinsically computationally demanding for high-dimensional problems (Yue et al., 2023) and difficult to tackle nonsmooth problems.

"RACOS" is a batch-mode classification-based DFO algorithm proposed by Yu et al. (2016). Compared with aforementioned algorithms, it shares the advantages of faster convergence rate, lower sensitivity to noise, available to high-dimensional problems and easy to implement (Qian et al., 2016a;b; Hu et al., 2017; Liu et al., 2017; 2019b). Additionally, "RACOS" has been proven to converge to global minimum in polynomial time when the objective is locally Holder continuous (Yu et al., 2016). The state-of-the-art classification-based DFO algorithm is "SRACOS" proposed by Hu et al. (2017), namely, the sequential-mode classification version. As proofs show, the sequential-mode classification version "SRACOS" performs better compared to the batch-mode counterpart "RACOS" under certain mild condition (more details can be found in Hu et al. (2017)).

However, "SRACOS" is a model-based algorithm whose convergence speed depends on the positive region of classification model, meaning that its convergent performance can be impacted by the dimensionality and measure of the solution space. Additionally, we find that the upper bound given in Yu et al. (2016); Hu et al. (2017) cannot completely dominate the complexity of "SRACOS". Moreover, we construct a counterexample (see in equation 3) showing that the upper bound on the other hand is not tight enough to describe the convergence speed.

We notice that a debatable assumption on the concept invoked by Yu et al. (2016) called error-target dependence (see Definition 2.2) takes charge of these minor deficiencies. In this paper we propose another concept called hypothesis-target $\eta$-shattering rate to replace the debatable assumption, and revisit the upper bound of computational complexity of the "SRACOS" (Hu et al., 2017). Inspired by the revisited upper bound, we propose a new classification-based DFO algorithm named "RACE-CARS" (the abbreviation of "RAndomized CoordinatE Classifying And Region Shrinking"), which inherits the characterizations of "SRACOS" while achieves acceleration theoretically. At last, we design experiments on the synthetic functions and black-box tuning for language-model-as-a-service comparing "RACE-CARS" with some state-of-the-art DFO algorithms and sequential "RACOS", illustrating the superiority of "RACE-CARS" empirically. In discussion, we empirically show (also under theoretical support) the ability of "RACE-CARS" beyond continuity. In addition, ablation experiment is performed to shed light on hyperparameter selection of the proposed algorithm.

The rest of the paper is organized in five sections, sequentially presenting the background, theoretical study, experiments, discussion and conclusion.

## 2 BACKGROUND

Let $\Omega$ be the solution space in $\mathbb{R}^n$, we presume that $\Omega$ is an $n$ dimensional compact cubic. In this work, we mainly focus on optimization problems reading as

$$\min_{x \in \Omega} f(x), \tag{1}$$

where only zeroth-order information is accessible for us, namely, once we input a potential solution $x \in \Omega$ into the oracle, merely the objective value $f(x)$ will return. In addition, we will not stipulate any convexity, smoothness or separability assumptions on $f$.

Assume $f(x)$ is lower bounded in $\Omega$ and $f^* := \min_{x \in \Omega} f(x)$. For the sake of theoretical analysis, we make some blanket notations: Denoted by $\mathcal{F}$ the Borel $\sigma$-algebra defined on $\Omega$ and $\mathbb{P}$ the probability measure on $\mathcal{F}$. For instance, when $\Omega$ is continuous, $\mathbb{P}$ is induced by Lebesgue measure $m$: $\mathbb{P}(B) := m(B)/m(\Omega)$ for all $B \in \mathcal{F}$. Let $\Omega_\epsilon := \{x \in \Omega : f(x) - f^* \le \epsilon\}$ for some $\epsilon > 0$. We always assume that for all $\epsilon > 0$, it holds $|\Omega_\epsilon| := \mathbb{P}(\Omega_\epsilon) > 0$.

A hypothesis (or classifier) $h$ is a function mapping the solution space $\Omega$ to $\{0, 1\}$. Define

$$D_h(x) := \begin{cases} 1/\mathbb{P}(\{x \in \Omega : h(x) = 1\}), & h(x) = 1 \\ 0, & \text{otherwise,} \end{cases} \tag{2}$$

a probability distribution in $\Omega$. Let $\mathbf{X}_h$ be the random vector in $(\Omega, \mathcal{F}, \mathbb{P})$ drawn from $D_h$, meaning that $\Pr(\mathbf{X}_h \in B) = \int_{x \in B} D_h(x) d\mathbb{P}$ for all $B \in \mathcal{F}$. Denoted by $\mathbb{T} := \{1, 2, \ldots, T\}$ and filtration $\mathbb{F} := (\mathcal{F}_t)_{t \in \mathbb{T}}$ a family of $\sigma$-algebras on $\Omega$ indexed by $\mathbb{T}$ such that $\mathcal{F}_1 \subseteq \mathcal{F}_2 \subseteq \cdots \subseteq \mathcal{F}_T \subseteq \mathcal{F}$. A typical classification-based optimization algorithm learns an $\mathbb{F}$-adapted stochastic process $\mathbf{X} :=$

$(\mathbf{X}_t)_{t \in \mathbb{T}}$, where $\mathbf{X}_t$ is induced by $\mathbf{X}_{h_t}$ and $h_t$ is the hypothesis learned at step $t$. Then samples new data with a stochastic process $\mathbf{Y} := (\mathbf{Y}_t)_{t \in \mathbb{T}}$ generated by $\mathbf{X}$ (Yu & Qian, 2014; Yu et al., 2016). Normally, the new solution at step $t \geq r + 1$ is sampled from

$$\mathbf{Y}_t := \begin{cases} \mathbf{X}_t, & \text{with probability } \lambda \\ \mathbf{X}_\Omega, & \text{with probability } 1 - \lambda, \end{cases}$$

where $\mathbf{X}_\Omega$ is random vector drawn from uniform sampling $\mathcal{U}_\Omega$ and $0 \leq \lambda \leq 1$ is the exploitation rate. A simplified batch-mode classification-based optimization algorithm is presented in appendix A Algorithm 2. At each step $t$, it selects a positive set $\mathcal{S}_{positive}$ from $\mathcal{S}$ containing the best $m$ samples, and the rest belong to negative set $\mathcal{S}_{negative}$. Then it trains a hypothesis $h_t$ which partitions the positive set and negative set such that $h_t(x_j) = 0$ for all $(x_j, y_j) \in \mathcal{S}_{negative}$. At last samples $r$ new solutions with the sampling random vector $\mathbf{Y}_t$. The sub-procedure $h_t \leftarrow \mathcal{T}(\mathcal{S}_{positive}, \mathcal{S}_{negative})$ trains a hypothesis under certain rules. "RACOS" is the abbreviation of "RAndomized COordinate Shrinking". Literally, it trains the hypothesis by this means (Yu et al., 2016), i.e., shrinking coordinates randomly such that all negative samples are excluded from positive region of resulting hypothesis. Algorithm 3 in appendix A shows a continuous version of "RACOS". The main difference between sequential-mode and batch-mode classification-based DFO is that the sequential version maintains a training set of size $r$ at each step $t$, then it samples only one solution after learning the hypothesis $h_t$. It replaces the training set with this new one under certain rules to finish step $t$. In the rest of this paper, we will omit the details of replacing sub-procedure which can be found in Hu et al. (2017). The pseudocode of sequential-model classification-based optimization algorithm is presented in Algorithm 4 in appendix A.

Classification-based DFO algorithms admit a performance bound on the query complexity, see Definition 2.1 (Yu & Qian, 2014), which counts the total number of calls to the objective function before finding a solution that reaches the approximation level $\epsilon$ with high probability $1 - \delta$. Definition 2.2 and 2.3 are given by Yu et al. (2016). The first one characterizes the so-called dependence between classification error and target region, which is expected to be as small as possible. The second one characterizes how large the positive region of hypothesis, which is also expected to be small.

**Definition 2.1** (($\epsilon, \delta$)-Query Complexity). Given $f, 0 < \delta < 1$ and $\epsilon > 0$, the ($\epsilon, \delta$)-query complexity of an algorithm $\mathcal{A}$ is the number of calls to $f$ such that, with probability at least $1 - \delta$, $\mathcal{A}$ finds at least one solution $\tilde{x} \in \Omega$ satisfying

$$f(\tilde{x}) - f^* \leq \epsilon.$$

**Definition 2.2** (Error-Target $\theta$-Dependence). The error-target dependence $\theta \geq 0$ of a classification-based optimization algorithm is its infimum such that, for any $\epsilon > 0$ and any $t$,

$$||\Omega_\epsilon| \cdot \mathbb{P}(\mathcal{R}_t) - \mathbb{P}(\Omega_\epsilon \cap \mathcal{R}_t)| \leq \theta |\Omega_\epsilon|,$$

where $\mathcal{R}_t := \Omega_{\alpha_t} \Delta \{x \in \Omega \colon h_t(x) = 1\}$ denotes the relative error, the operator $\Delta$ is the symmetric difference of two sets defined as $A_1 \Delta A_2 = (A_1 \cup A_2) - (A_1 \cap A_2)$. Similar to the definition of $\Omega_\epsilon$, $\Omega_{\alpha_t} := \{x \in \Omega \colon f(x) - f^* \leq \alpha_t\}$ with $\alpha_t := \min_{1 \leq i \leq t} f(x_i) - f^*$.

**Definition 2.3** ($\gamma$-Shrinking Rate). The shrinking rate $\gamma > 0$ of a classification-based optimization algorithm is its infimum such that $\mathbb{P}(x \in \Omega \colon h_t(x) = 1) \leq \gamma |\Omega_{\alpha_t}|$ for all $t$.

**Theorem 2.1.** (Hu et al., 2017) Given $0 < \delta < 1$ and $\epsilon > 0$, if a sequential classification-based optimization algorithm has error-target $\theta$-dependence and $\gamma$-shrinking rate, then its ($\epsilon, \delta$)-query complexity is upper bounded by

$$\mathcal{O}\left(\max\left\{\frac{1}{|\Omega_\epsilon|}\left(\lambda + \frac{1-\lambda}{\gamma(T-r)}\sum_{t=r+1}^{T}\Phi_t\right)^{-1}\ln\frac{1}{\delta}, T\right\}\right),$$

where the $\Phi_t = \left(1 - \theta - \mathbb{P}(\mathcal{R}_{D_t}) - m(\Omega)\sqrt{\frac{1}{2}D_{\mathrm{KL}}(D_t\|\mathcal{U}_\Omega)}\right) \cdot |\Omega_{\alpha_t}|^{-1}$ with the notations $D_t := \lambda D_{h_t} + (1-\lambda)\mathcal{U}_\Omega$ and $\mathbb{P}(\mathcal{R}_{D_t}) := \int_{\mathcal{R}_t} D_t d\mathbb{P}$.

# 3 THEORETICAL STUDY

## 3.1 DEFICIENCIES CAUSED BY ERROR-TARGET DEPENDENCE

**(i) The query complexity of Algorithm 4 cannot be upper-bounded by Theorem 2.1.**

On the basis of error-target $\theta$-dependence and $\gamma$-shrinking rate, Theorem 2.1 gives a general bound of the query complexity of the sequential-mode classification-based DFO algorithm 4. As assumptions entailed in Yu et al. (2016); Hu et al. (2017), it can be easily observed that the smaller $\theta$ or $\gamma$ the better the query complexity. However, in some cases even though these two factors are small, something wrong happens. Following the lemma given by Yu et al. (2016): $\mathbb{P}(\mathcal{R}_t) \leq \mathbb{P}(\mathcal{R}_{D_t}) + m(\Omega)\sqrt{\frac{1}{2}D_{\mathrm{KL}}(D_t\|\mathcal{U}_\Omega)}$, it holds

$$\Phi_t = \left(1 - \theta - \mathbb{P}(\mathcal{R}_{D_t}) - m(\Omega)\sqrt{\frac{1}{2}D_{\mathrm{KL}}(D_t\|\mathcal{U}_\Omega)}\right) \cdot |\Omega_{\alpha_t}|^{-1} \leq (1 - \theta - \mathbb{P}(\mathcal{R}_t)) \cdot |\Omega_{\alpha_t}|^{-1}.$$

According to the definition of error-target $\theta$-dependence, small $\theta$ does not indicate small relative error $\mathbb{P}(\mathcal{R}_t)$. Contrarily, small $\theta$ and big $(\mathbb{P}(\Omega_\epsilon \cap \mathcal{R}_t))/|\Omega_\epsilon|$ implies big $\mathbb{P}(\mathcal{R}_t)$, which can even be 1 as long as $\Omega_{\alpha_t}$ is totally out of the positive region of $h_t$, namely the situation that hypothesis $h_t$ is 100% wrong. Then $\Phi_t \leq (1 - \theta - \mathbb{P}(\mathcal{R}_t)) \cdot |\Omega_{\alpha_t}|^{-1} \leq 0$, which is unacceptable since $\Phi_t$ performs as a divisor in the proof of Theorem 2.1, the negative divisor will change the sign of inequality relationship. Actually, in order that $\Phi_t \leq 0$, $\mathbb{P}(\mathcal{R}_t)$ is not necessary to be 1 since $\theta$ is nonnegative. In other words, a sequence of inaccurate hypotheses suffice to make the upper bound wrong.

**(ii)** **The upper bound is not tight enough.**

Let us consider an extreme situation. Assume that the hypotheses at each step are all

$$h_t(x) = \begin{cases} 1, & x \in \Omega_\epsilon \\ 0, & x \notin \Omega_\epsilon. \end{cases} \tag{3}$$

Since the size of training sets used to train the hypothesis $h_t$ is tiny and the training data is biased in the context of sequential-mode classification-based optimization algorithm, it is quite reasonable to assume that the relative errors $\mathbb{P}(\mathcal{R}_t)$ are large. In other words, in this situation we learn a series of "inaccurate" hypotheses with respect to $\Omega_{\alpha_t}$ while accidentally "accurate" with respect to $\Omega_\epsilon$. Consequently, the error-target dependence $\theta = \max_{1 \leq t \leq T} \mathbb{P}(\mathcal{R}_t)$ is large. Even though $\Phi_t$ are all positive, the query complexity bound given in Theorem 2.1 can be large. However, in this situation, it can be easily proven that the probability an $\epsilon$-minimum is not found until step $T$ is

$$\Pr\left(\min_{1 \leq t \leq T} f(x_t) - f^* \geq \epsilon\right) = (1 - |\Omega_\epsilon|)^r \left((1 - \lambda)(1 - |\Omega_\epsilon|)\right)^{T-r},$$

which is smaller than $\delta$ for not very large $T$. It means that the upper bound in Theorem 2.1 is not tight.

## 3.2 REVISIT OF QUERY COMPLEXITY UPPER BOUND

Considering the minor deficiencies caused by error-target dependence, it can be observed that regardless how small error-target dependence is, deficiencies happen due to large relative error of classification, since error-target dependence cannot dominate relative error. It seems that an assumption on small relative error should be supplemented. However, this kind of assumption is not reasonable since the size of training sets is tiny and the training data is biased in the context of sequential-mode classification-based optimization algorithm. To this end, we give a new concept that is independent to the influence of relative error:

**Definition 3.1** (Hypothesis-Target $\eta$-Shattering Rate). Given $\eta \in [0,1]$, for a family of hypotheses $\mathcal{H}$ defined on $\Omega$, we say $\Omega_\epsilon$ is $\eta$-shattered by $h \in \mathcal{H}$ if

$$\mathbb{P}(\Omega_\epsilon \cap \{x \in \Omega : h(x) = 1\}) \geq \eta|\Omega_\epsilon|,$$

and $\eta$ is called hypothesis-target shattering rate.

Similar to error-target dependence, hypothesis-target shattering rate is relevant to certain accuracy of the hypothesis. Moreover, the error-target dependence can be bounded by relative error and hypothesis-target shattering rate: $\theta \leq \max\{\mathbb{P}(\mathcal{R}_t), |1 - \mathbb{P}(\mathcal{R}_t) - \eta|\}$. Hypothesis-target shattering rate $\eta$ describes how large the intersection of target $\Omega_\epsilon$ and positive region of hypothesis. Additionally, it eliminates the impact of relative error on error-target dependence. In the following theorem, we revisit the upper bound of $(\epsilon, \delta)$-query complexity with hypothesis-target shattering rate.

**Theorem 3.1.** Consider sequential-mode classification-based DFO Algorithm 4, let $\mathbf{X}_t = \mathbf{X}_{h_t}$, assume that for $\epsilon > 0$, $\Omega_\epsilon$ is $\eta$-shattered by $h_t$ for all $t = r + 1 \ldots, T$ and $\max\limits_{t=r+1,\ldots,T} \mathbb{P}(\{x \in \Omega \colon h_t(x) = 1\}) \leq p \leq 1$. Then for $0 < \delta < 1$, the $(\epsilon, \delta)$-query complexity is upper bounded by

$$\mathcal{O}\left( \max\{(\lambda\frac{\eta}{p} + (1 - \lambda))^{-1}\left(\frac{1}{|\Omega_\epsilon|} \ln \frac{1}{\delta} - r\right) + r, T\} \right).$$

The detailed proof of Theorem 3.1 can be found in appendix F.1.

### 3.3 An Acceleration for Sequential-Mode Classification-Based DFO

Actually, in terms of the $(\epsilon, \delta)$-query complexity analysis of classification-based optimization algorithm, the relative error $\mathbb{P}(\mathcal{R}_t)$ does not play a decisive role since we aim at finding the optimum rather than learning a series of accurate classifier. As the counterexample equation 3 which is literally the most desirable hypothesis although the relative error can be large, we should focus more on the intersection of $\Omega_\epsilon$ and positive region of hypotheses, namely the hypothesis-target shattering rate.

Shrinking rate defined in Definition 2.3 describes the decaying speed of $\mathbb{P}(x \in \Omega \colon h_t(x) = 1)$. However, on the one hand, training a well-fitted hypothesis to $\Omega_{\alpha_t}$ is not our original intention; On the other hand $|\Omega_{\alpha_t}|$ decreases rapidly as $\alpha_t \to 0$, whereas it is also unrealistic to maintain such a series of small $\gamma$-shrinking hypotheses by means of sequential randomized coordinate shrinking method. Therefore, $\gamma$-shrinking assumption is usually not guaranteed for small $\gamma$.

Instead of pursuing small relative error and $\gamma$-shrinking with respect to $|\Omega_{\alpha_t}|$, we propose the following Algorithm 1 "sequential RAndomized CoordinatE Classifying And Region Shrinking" (sequential "RACE-CARS"), which shrink the positive region of the sampling random vector $\mathbf{Y}_t$ proactively and adaptively via a projection sub-procedure. We put a 2-dimensional illustration as follows:

The left subfigure is the intersection before projection, where the ellipse centered on $x^*$ is the intersection of $\Omega_\epsilon$ and the rectangle containing $x_{best}$ is the positive region of $h_t$ such that $h_t(x) = 1$. The right one shows the intersection after projection, where the positive region of $h_t$ shrinks to the positive region. 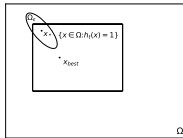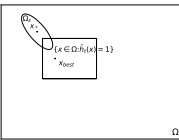

---

**Algorithm 1:** Sequential Randomized Coordinate Classifying and Region Shrinking Algorithm

**Input:**
  $\Omega$: Boundary;  $T \in \mathbb{N}^+$: Budget;  $r = m + k$;  $Replace$: Replacing sub-procedure;
  $\gamma$: Region shrinking rate;  $\rho$: Region shrinking frequency.
**Output:** $(x_{best}, y_{best})$.

Collect $\mathcal{S} = \{(x_1, y_1), \ldots, (x_r, y_r)\}$ i.i.d. from $\mathcal{U}_\Omega$;
$(x_{best}, y_{best}) = \arg\min\{y \colon (x, y) \in \mathcal{S}\}$;
Initialize $k = 1, \tilde{\Omega} = \Omega$;
**for** $t = r + 1, \ldots, T$ **do**
  Train: $h_t \leftarrow \mathcal{T}(\mathcal{S}_{positive}, \mathcal{S}_{negative})$ through "RACOS";
  s $\leftarrow random(0, 1)$;
  **if** s $\leq \rho$ **then**
    Shrink region: $\tilde{\Omega} = [x_{best} - \frac{1}{2}\gamma^k\|\Omega\|, x_{best} + \frac{1}{2}\gamma^k\|\Omega\|] \cap \Omega$;
    $k = k + 1$;
  Project: $\mathbf{Y}_t \leftarrow Proj(h_t, \tilde{\Omega})$;
  Sample: $(x_t, y_t) \sim \mathbf{Y}_t$;
  Replace: $\mathcal{S} \leftarrow Replace((x_t, y_t), \mathcal{S})$;
  $(x_{best}, y_{best}) = \arg\min\{y \colon (x, y) \in \mathcal{S}\}$;
**return** $(x_{best}, y_{best})$

---

The operator $\|\cdot\|$ returns a tuple comprised of the diameter of each dimension of the region. For instance, when $\Omega = [\omega_1^1, \omega_2^1] \times [\omega_1^2, \omega_2^2]$, we have $\|\Omega\| = (\omega_2^1 - \omega_1^1, \omega_2^2 - \omega_1^2)$. The projection operator

$Proj(h_t, \tilde{\Omega})$ generates a random vector $\mathbf{X}_t$ with probability distribution

$$D_{\tilde{h}_t} := \tilde{h}_t / \mathbb{P}(\{x \in \Omega \colon \tilde{h}_t(x) = 1\}),$$

where $\tilde{h}_t(x) = 1$ when $h_t(x) = 1$ and $x \in \tilde{\Omega}$. The sampling random vector $\mathbf{Y}_t$ is induced from $\mathbf{X}_t$. The following theorem gives the query complexity upper bound of Algorithm 1.

**Theorem 3.2.** Consider Algorithm 1. Assume that for $\epsilon > 0$, $\Omega_\epsilon$ is $\eta$-shattered by $\tilde{h}_t$ for all $t = r+1 \ldots, T$. Let the region shrinking rate $0 < \gamma < 1$ and region shrinking frequency $0 < \rho < 1$, then for $0 < \delta < 1$, the $(\epsilon, \delta)$-query complexity of "RACE-CARS" is upper bounded by

$$\mathcal{O}\left( \max\{ \left( \frac{\gamma^{-\rho} + \gamma^{-(T-r)\rho}}{2} \lambda\eta + (1-\lambda) \right)^{-1} \left( \frac{1}{|\Omega_\epsilon|} \ln \frac{1}{\delta} - r \right) + r, T \} \right).$$

The detailed proof of Theorem 3.2 can be found in appendix F.2.

$0 < \gamma < 1$ implies $\frac{2p}{\gamma^{-\rho} + \gamma^{-(T-r)\rho}} \ll 1$. Theorem 3.2 indicates that the upper bound of the $(\epsilon, \delta)$-query complexity of sequential "RACE-CARS" is smaller than "SRACOS" as long as $\eta > 0$.

**Definition 3.2** (Dimensionally local Holder continuity). Assume that $x_* = (x_*^1, \ldots, x_*^n)$ is the unique global minimum such that $f(x_*) = f^*$. We call $f$ dimensionally local Holder continuity if

$$L_1^i |x^i - x_*^i|^{\beta_1^i} \leq |f(x_*^1, \ldots, x^i, \ldots, x_*^n) - f^*| \leq L_2^i |x^i - x_*^i|^{\beta_2^i}, \quad \forall i = 1, \ldots, n,$$

for all $x = (x^1, \ldots, x^n)$ in the neighborhood of $X_*$, where $\beta_1^i, \beta_2^i, L_1^i, L_2^i$ are positive constants for $i = 1, \ldots, n$.

Under the assumption that $f$ is dimensionally locally Holder continuous, it is obvious that

$$\Omega_\epsilon \subseteq \prod_{i=1}^n [x_*^i - \left(\frac{\epsilon}{L_1^i}\right)^{-\beta_1^i}, x_*^i + \left(\frac{\epsilon}{L_1^i}\right)^{-\beta_1^i}].$$

Denoted by $\tilde{x}_t = (\tilde{x}_t^1, \ldots, \tilde{x}_t^n) := \arg\min_{j=1,\ldots,t} f(x_j)$, the following theorem gives a lower bound of region shrinking rate $\gamma$ and shrinking frequency $\rho$, such that "RACE-CARS" meets the assumptions in Theorem 3.2.

**Theorem 3.3.** For a dimensionally local Holder continuous objective $f$. Assume that for $\epsilon > 0$, $\Omega_\epsilon$ is $\eta$-shattered by $h_t$ for all $t = r+1, \ldots, T$. In order that $\Omega_\epsilon$ being $\eta$-shattered by $\tilde{h}_t$, it is sufficient when the region shrinking rate $\gamma$ and shrinking frequency $\rho$ satisfy:

$$\frac{1}{2}\gamma^{t\rho}\|\Omega\| \geq \left( \tilde{x}_t^1 - x_*^1 + \left(\frac{\epsilon}{L_1^1}\right)^{-\beta_1^1}, \ldots, \tilde{x}_t^n - x_*^n + \left(\frac{\epsilon}{L_1^n}\right)^{-\beta_1^n} \right).$$

However, $\beta_1^i, L_1^i$ and $x_*$ are unknown generally, $\gamma$ and $\rho$ are hyperparameters should be carefully selected. Although new hyperparameters are introduced, they make practical sense and there are traces to follow when tuning them. Empirical hyperparameter-tuning guidance is conducted in section 5.

## 4 EXPERIMENTS

In this section, we design 2 experiments to test "RACE-CARS" on synthetic functions, and a language model task respectively. We use same budget to compare "RACE-CARS" with several state-of-the-art DFO algorithms, including sequential "RACOS" ("SRACOS") (Hu et al., 2017), zeroth-order adaptive momentum method ("ZO-Adam") (Chen et al., 2019), differential evolution ("DE") (Opara & Arabas, 2019) and covariance matrix adaptation evolution strategies ("CMA-ES") (Hansen, 2016). All the baseline algorithms are fine-tuned.

### 4.1 ON SYNTHETIC FUNCTIONS

We first test on four well-known benchmark test functions: Ackley, Levy, Rastrigin and Sphere. Analytic expressions can be found in appendix B and Figure 3 shows the surfaces of four 2-dimensional test functions. It can be observed that they are highly nonconvex with many local minima except

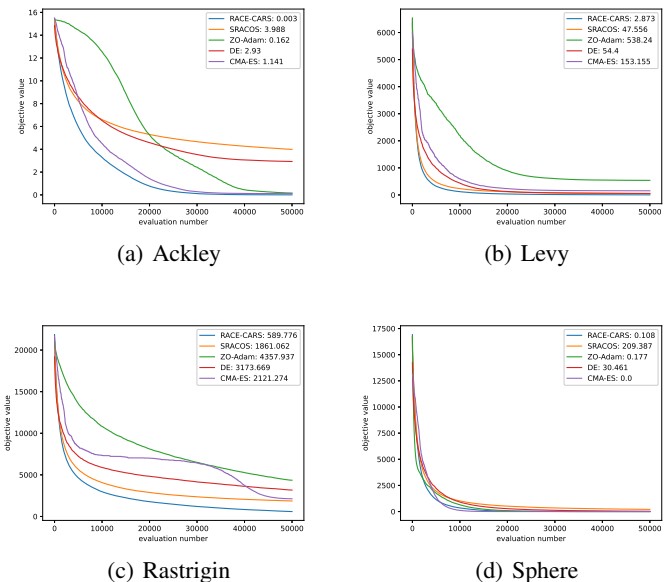

**Figure 1: Comparison of the synthetic functions with n = 500.**

for Sphere. These functions are minimized within the boundary $\Omega = [-10, 10]^n$, and the minimum of them are all 0. We choose the dimension of solution space $n$ to be 50 and 500, and the budget of function evaluation is set to be 5000 and 50000 respectively (as the results show, in order to make the algorithm converge, only a few portions of the budget is enough for "RACE-CARS"). Region shrinking rate is set to be $\gamma = 0.9, 0.95$ and region shrinking frequency is $\rho = 0.01, 0.001$ respectively when $n = 50, 500$. Each of the algorithm is repeated 30 runs and the convergence trajectories of mean of the best-so-far value are presented in Figure 4 (in appendix C) and Figure 1. The numbers attached to the algorithm names in the legend of figures are the mean value of obtained minimum. It can be observed that "RACE-CARS" performs the best on both convergence speed and optimal value, except for the strongly convex function Sphere, where it is slightly worse than "CMA-ES". However, it should be emphasized that "CMA-ES" involves an $n$-dimensional covariance matrix, which is very time-consuming and suffers from scalability issue compared with the other four algorithms.

## 4.2 On Black-Box Tuning for Language-Model-as-a-Service

Prompt tuning for extremely large pre-trained language models (PTMs) has shown great power. PTMs such as GPT-3 (Brown et al., 2020) are usually released as a service due to commercial considerations and the potential risk of misuse, allowing users to design individual prompts to query the PTMs through black-box APIs. This scenario is called Language-Model-as-a-Service (LMaaS) (Sun et al., 2022; Diao et al., 2022). In this part we follow the experiments designed by Sun et al. (2022) [1], where language understanding task is formulated as a classification task predicting for a batch of PTM-modified input texts $X$ the labels $Y$ in the PTM vocabulary, namely we need to tune the prompt such that the black-box PTM inference API $f$ takes a continuous prompt $\mathbf{p}$ satisfying $Y = f(\mathbf{p}; X)$. Moreover, to handle the high-dimensional prompt $\mathbf{p}$, Sun et al. (2022) proposed to randomly embed the $D$-dimensional prompt $\mathbf{p}$ into a lower dimensional space $\mathbb{R}^d$ via random projection matrix $\boldsymbol{A} \in \mathbb{R}^{D \times d}$. Therefore, the objective becomes:

$$\min_{z \in \mathcal{Z}} \mathcal{L}\big(f(\boldsymbol{A}z + \mathbf{p}_0; X), Y\big),$$

where $\mathcal{Z} = [-50, 50]^d$ is the search space and $\mathcal{L}(\cdot)$ is cross entropy loss. In our experiments, we set the dimension of search space as $d = 500$, prompt length as 50, let RoBERTa (Liu et al., 2019a)

---

[1]Code can be found in https://github.com/txsun1997/Black-Box-Tuning

be backbone model. We test on datasets SST-2 (Socher et al., 2013), Yelp polarity and AG's News (Zhang et al., 2015), and RTE (Wang et al., 2018). Under the same budget of API calls $T = 8000$, we compare "RACE-CARS" with "SRACOS" and the default DFO algorithm "CMA-ES" employed in Sun et al. (2022) [2]. The shrinking rate is $\gamma = 0.7$ and shrinking frequency is $\rho = 0.002$. Each of the algorithm is repeated 5 runs independently with different seeds. Figure 2 shows comparisons on the performance of mean and deviation of training loss. In appendix C, results on training accuracy (Figure 5), development loss (Figure 6) and development accuracy (Figure 7) can be found. Results show that "RACE-CARS" accelerates "SRACOS" in each case. "CMA-ES" outperforms "RACE-CARS" on Yelp polarity, AG's News and RTE, while it is a bit overfitting compared to "RACE-CARS" (see Figure 6 and 7). "RACE-CARS" realizes comparable performance to "CMA-ES". In contrast to the well-tuned "CMA-ES", the hyperparameters of "RACE-CARS" are only tuned empirically w.r.t. SST-2 dataset, and then directly applied on three other datasets.

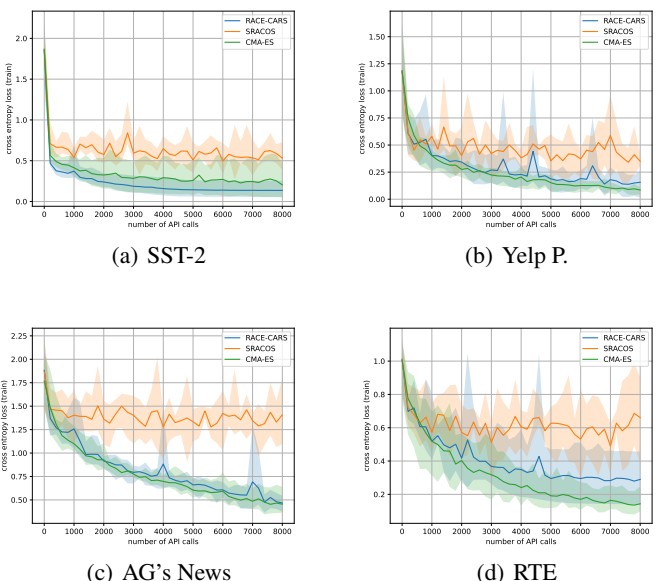

(a) SST-2    (b) Yelp P.

(c) AG's News    (d) RTE

Figure 2: Comparison of black-box tuning for LMaaS (Training loss).

## 5    DISCUSSION

### 5.1    BEYOND CONTINUITY

(i) For discontinuous objective functions.

The sufficient condition Theorem 3.3 gives a hyperparameters selection strategy when the objective is dimensionally local Holder continuous (see Definition 3.2), in which the objective is restricted by several continuous envelopes whereas not supposed to be continuous. Aside from continuous cases in the last section, when dealing with discontinuous objectives, "RACE-CARS" is still valid. See appendix E for details.

(ii) For discrete optimization.

Similar to "SRACOS", "RACE-CARS" maintains the ability to solve discrete optimization problems. Theorem 3.1 and 3.2 can fit this situation when altering the measure of probability space to be for example, induced by counting measure. We further conduct experiments on mixed-integer programming problems, showing the acceleration of "RACE-CARS" empirically. See appendix E for details.

---

[2] As section 4.1 shows, neither "ZO-Adam" nor "DE" achieves satisfactory result when the objective is a high-dimensional nonconvex black-box function. In the consideration of concision and explicitation, we discard these two methods in this section.

## 5.2 Ablation Experiments

Generally, the word "black-box" implies objective-agnosticism or partial-cognition. Just like what "No Free Lunch" theorem tells, it is desirable but not realistic to design a universal well-performed DFO algorithm for black-box functions while hyperparameter-free. We propose "RACE-CARS" carrying 2 hyperparameters shrinking-rate $\gamma$ and shrinking-frequency $\rho$. For an $n$-dimensional optimization problem, we call $\gamma^{n\rho}$ shrinking factor of "RACE-CARS". In Theorem 3.3 we give a lower bound of shrinking factor in the expectation sense. In this subsection we take Ackley for a case study, design ablation experiments on the 2 hyperparameters of "RACE-CARS" to reveal the mechanism. We stipulate that we do not aim to find a best composition of hyperparameters, whereas to put forth an empirical hyperparameter-tuning guidance. See appendix E for details.

## 5.3 On the Newly Introduced Assumption

In the former studies, Hu et al. (2017) establish query complexity upper bound based on two quantities, namely the error-target $\theta$-dependence (see Definition 2.2) and the $\gamma$-shrinking rate (see Definition 2.3). According to the analysis in subsection 3.1 and the second paragraph in subsection 3.3, these two quantities may lead to inaccurate upper bound under certain extreme conditions. The pertinence of these two quantities is therefore debatable. In the present study, we deprecate these quantities and introduce a more straightforward concept, i.e. "hypothesis-target $\eta$-shattering" (see Definition 3.1), which describes the percentage $\eta$ of the intersection of hypotheses and target. According to the derived Theorem 3.1 and Theorem 3.2, region-shrinking accelerates the classification-based DFO as long as $\eta > 0$. Following the idea of this newly introduced assumption, training a series of accurate hypotheses is not necessary, while pursuing bigger shattering rate $\eta$ is more preferred. This *radical* strategy is in contrast to Hashimoto et al. (2018), who propose a batch-mode classification-based DFO using a *conservative Training* sub-procedure, namely consistent selective strategy (CSS) (El-Yaniv & Wiener, 2012). They prove that for a hypothesis class $\mathcal{H}$ of VC-dimension $VC(\mathcal{H})$, an accurate hypothesis needs $\mathcal{O}(\epsilon^{-1}(VC(\mathcal{H})\log(\epsilon^{-1})))$ per-batch samples for *Training*. Although they propose a computationally efficient approximation, only successes on low dimensional problems are provided and performance on high dimensional remains unclear.

However, it is not trivial to identify the applicable scope of the newly introduced assumption, given the complexity of the stochastic process corresponding to the *Training* and *Replacing* sub-procedure (see equation 2, Algorithm 3 and Algorithm 4). To our best knowledge, the conditional expectation of such process cannot be determined analytically. We plan to address this gap via two routes in our future work. Firstly, despite the provided empirical test results, numerical tools compromising black-box function exploration and computational cost may be useful to further clarify the applicability of the proposed algorithm. Furthermore, altering the *Training* and *Replacing* sub-procedures inherited from "RACOS", which may ideally lead to a bigger shattering rate and maintain the easy-to-sample characterization, will be another extension direction of the current study.

## 6 Conclusion

In this paper, we propose a concept called hypothesis-target shattering rate and revisit the query complexity upper bound of sequential-mode classification-based DFO algorithms:

$$\mathcal{O}\bigg(\max\{\big(\lambda\frac{\eta}{p} + (1-\lambda)\big)^{-1}\big(\frac{1}{|\Omega_\epsilon|}\ln\frac{1}{\delta} - r\big) + r, T\}\bigg).$$

Inspired by the computational complexity upper bound under the new framework, we propose a region-shrinking technique to accelerate the convergence. Computational complexity upper bound of the derived sequential classification-based DFO algorithm "RACE-CARS" is:

$$\mathcal{O}\bigg(\max\{\big(\frac{\gamma^{-\rho} + \gamma^{-(T-r)\rho}}{2}\lambda\eta + (1-\lambda)\big)^{-1}\big(\frac{1}{|\Omega_\epsilon|}\ln\frac{1}{\delta} - r\big) + r, T\}\bigg).$$

The 2 newly introduced hyperparameters $\gamma \in (0, 1)$ is shrinking rate and $\rho \in (0, 1)$ is shrinking frequency. Since $\frac{2p}{\gamma^{-\rho}+\gamma^{-(T-r)\rho}} \ll 1$, "RACE-CARS" outperforms "SRACOS" theoretically. In empirical analysis, we study the performance of "RACE-CARS" on synthetic functions and black-box tuning for language-model-as-a-service, showing its superiority over "SRACOS" and other 4 DFO algorithms.

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

# A ALGORITHMS

---

**Algorithm 2:** Batch-Mode Classification-Based Optimization Algorithm

---

**Input:**
$T \in \mathbb{N}^+$: Budget;    $r$: Training size;    $m$: Positive size.
**Output:** $(x_{best}, y_{best})$.

Collect: $\mathcal{S} = \{(x_1^0, y_1^0), \ldots, (x_r^0, y_r^0)\}$ i.i.d. from $\mathcal{U}_\Omega$;
$(x_{best}, y_{best}) = \arg\min\{y \colon (x, y) \in \mathcal{S}\}$;
**for** $t = 1, \ldots, T/r$ **do**
    Classify: $(\mathcal{S}_{positive}, \mathcal{S}_{negative}) \leftarrow \mathcal{S}$;
    Train: $h_t \leftarrow \mathcal{T}(\mathcal{S}_{positive}, \mathcal{S}_{negative})$;
    Sample: $\{(x_1^t, y_1^t), \ldots, (x_r^t, y_r^t)\}$ i.i.d. with $\mathbf{Y}_t$;
    Select: $\mathcal{S} \leftarrow$ best $r$ samples within $\mathcal{S} \cup \{(x_1^t, y_1^t), \ldots, (x_r^t, y_r^t)\}$;
    $(x_{best}, y_{best}) = \arg\min\{y \colon (x, y) \in \mathcal{S}\}$.
**return** $(x_{best}, y_{best})$

---

**Algorithm 3:** RACOS

---

**Input:**
$\Omega$: Boundary;    $(\mathcal{S}_{positive}, \mathcal{S}_{negative})$: Binary sets;    $\mathbb{I} = \{1, \ldots, n\}$: Index of dimensions.
**Output:** $h$: Hypothesis.

Randomly select: $x_+ = (x_+^1, \ldots, x_+^n) \leftarrow \mathcal{S}_{positive}$;
$h(x) \equiv 1$;
**while** $\exists x \in \mathcal{S}_{negative}$ *s.t.* $h_t(x) = 1$ **do**
    Randomly select: $k \leftarrow \mathbb{I}$;
    Randomly select: $x_- = (x_-^1, \ldots, x_-^n) \leftarrow \mathcal{S}_{negative}$;
    **if** $x_+^k \leq x_-^k$ **then**
        $s \leftarrow random(x_+^k, x_-^k)$;
        Shrink: $h(x) = 0, \forall x \in \{x = (x^1, \ldots, x^n) \in \Omega \colon x^k > s\}$;
    **else**
        $s \leftarrow random(x_-^k, x_+^k)$;
        Shrink: $h(x) = 0, \forall x \in \{x = (x^1, \ldots, x^n) \in \Omega \colon x^k < s\}$;
    **return** $h$

---

**Algorithm 4:** Sequential-Mode Classification-Based Optimization Algorithm

---

**Input:**
$T \in \mathbb{N}^+$: Budget;    $r$: Training size;    $m$: Positive size;
$Replace$: Replacing sub-procedure.
**Output:** $(x_{best}, y_{best})$.

Collect $\mathcal{S} = \{(x_1, y_1), \ldots, (x_r, y_r)\}$ i.i.d. from $\mathcal{U}_\Omega$;
$(x_{best}, y_{best}) = \arg\min\{y \colon (x, y) \in \mathcal{S}\}$;
**for** $t = r + 1, \ldots, T$ **do**
    Classify: $(\mathcal{S}_{positive}, \mathcal{S}_{negative}) \leftarrow \mathcal{S}$;
    Train: $h_t \leftarrow \mathcal{T}(\mathcal{S}_{positive}, \mathcal{S}_{negative})$;
    Sample: $(x_t, y_t) \sim \mathbf{Y}_t$;
    Replace: $\mathcal{S} \leftarrow Replace((x_t, y_t), \mathcal{S})$;
    $(x_{best}, y_{best}) = \arg\min\{y \colon (x, y) \in \mathcal{S}\}$;
**return** $(x_{best}, y_{best})$

---

# B    SYNTHETIC FUNCTIONS

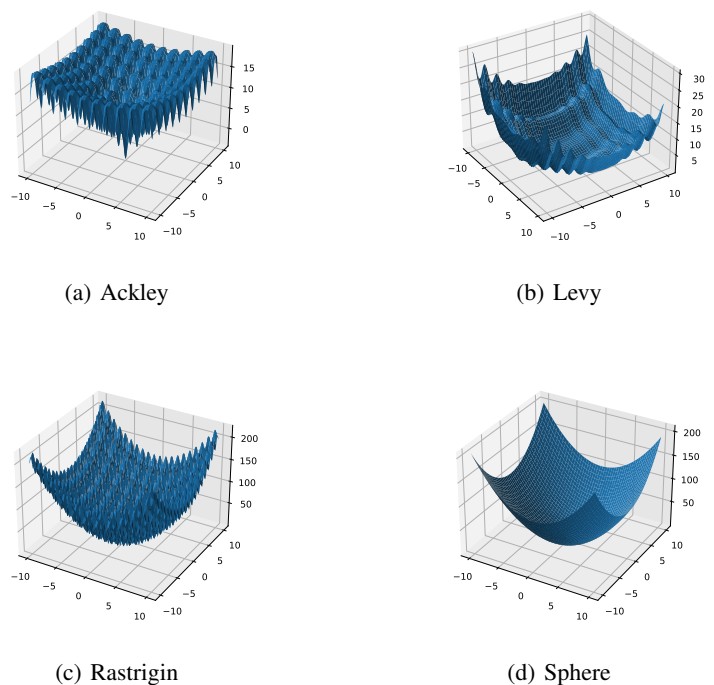

(a) Ackley

(b) Levy

(c) Rastrigin

(d) Sphere

**Figure 3: Synthetic functions with n = 2.**

- Ackley:

$$f(x) = -20 \exp(-0.2\sqrt{\sum_{i=1}^{n}(x_i - 0.2)^2/n}) - \exp(\sum_{i=1}^{n}\cos 2\pi x_i/n) + e + 20.$$

- Levy:

$$f(x) = \sin^2(\pi\omega_1) + \sum_{i=1}^{n-1}(\omega_i - 1)^2\big(1 + 10\sin^2(\pi\omega_i + 1)\big) + (\omega_n - 1)^2\big(1 + \sin^2(2\pi\omega_n)\big)$$

where $\omega_i = 1 + \frac{x_i - 1}{4}$.

- Rastrigin:

$$f(x) = 10n + \sum_{i=1}^{n}\big(x_i^2 - 10\cos(2\pi x_i)\big).$$

- Sphere:

$$f(x) = \sum_{i=1}^{n}(x_i - 0.2)^2.$$

# C FIGURES

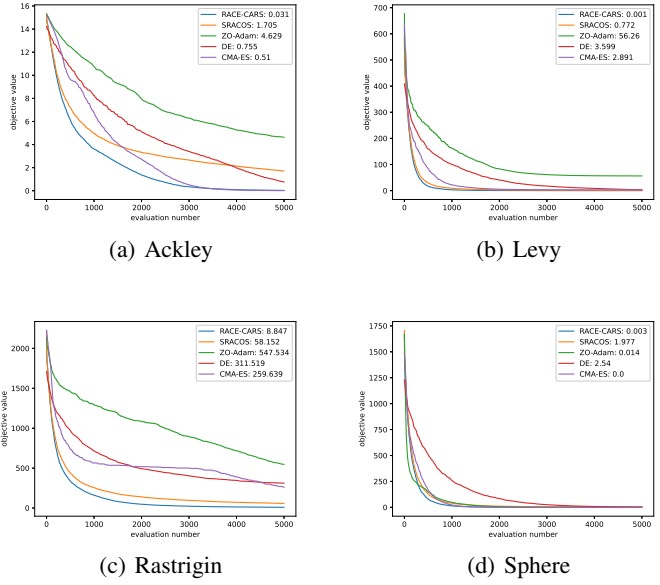

Figure 4: **Comparison of the synthetic functions with** $n = 50$.

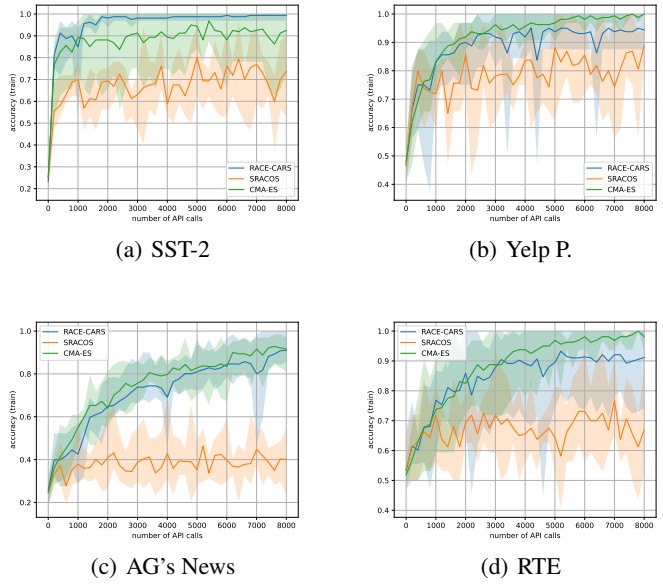

Figure 5: **Comparison of black-box tuning for LMaaS (Training accuracy).**

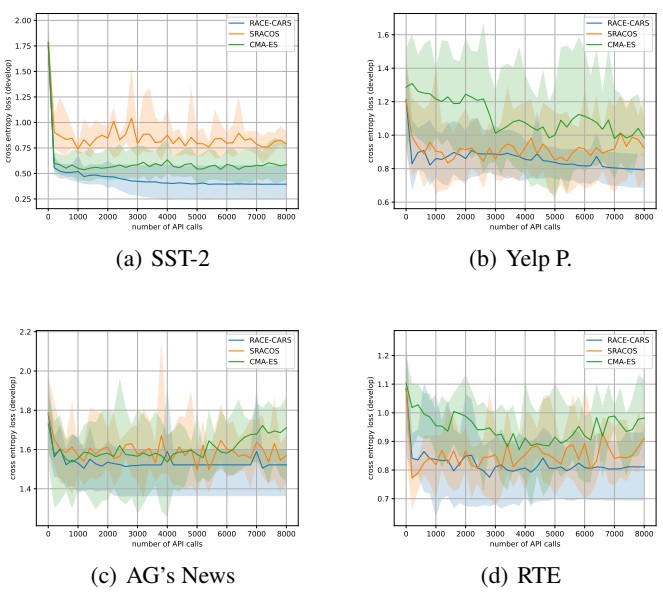

(a) SST-2          (b) Yelp P.

(c) AG's News        (d) RTE

Figure 6: **Comparison of black-box tuning for LMaaS (Development loss).**

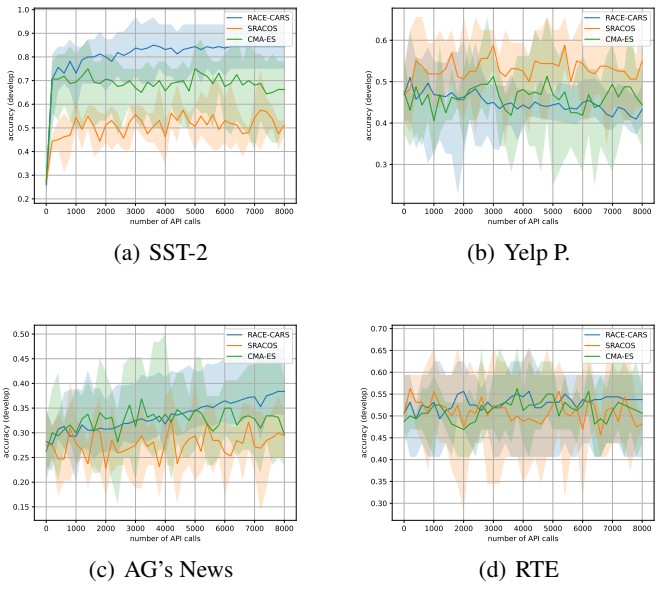

(a) SST-2          (b) Yelp P.

(c) AG's News        (d) RTE

Figure 7: **Comparison of black-box tuning for LMaaS (Development accuracy).**

# D  TABLES

**Table 1: Comparison of shrinking frequencies $\rho$ for Ackley on $\Omega = [-10, 10]^n$ with shrinking rate $\gamma = 0.95$.** Mean and standard deviation of function value at the $30n$ step are listed in the table. The first row of the table with $\rho = 0$ is the results of "SRACOS" for reference. We omit the results of $\rho$ bigger than 0.1 for concision. The bold fonts are relative better results in each dimension.

| $\rho$ \ n | 50 | 100 | 150 | 200 | 250 | 300 | 350 | 400 | 450 | 500 |
|---|---|---|---|---|---|---|---|---|---|---|
| **0** | 3.8±0.2 | 3.9±0.2 | 5.9±0.1 | 5.7±0.1 | 5.8±0.2 | 5.8±0.1 | 5.9±0.0 | 5.8±0.0 | 5.9±0.1 | 5.8±0.1 |
| 0.002 | 3.7±0.2 | 3.5±0.2 | 4.3±0.3 | 4.4±0.3 | 4.0±0.6 | 3.9±0.5 | 3.7±0.4 | 3.3±0.4 | 3.3±0.3 | 2.6±0.4 |
| 0.004 | 3.4±0.3 | 3.2±0.1 | 3.8±0.6 | 3.7±0.2 | 2.8±0.5 | 2.1±0.5 | **1.9±0.2** | **1.9±0.2** | **1.8±0.6** | **1.7±0.4** |
| 0.006 | 3.3±0.3 | 2.9±0.2 | 2.9±0.4 | 2.0±0.1 | 2.1±0.4 | **1.8±0.2** | **1.9±0.2** | 2.4±0.3 | 2.7±0.3 | 3.1±1.5 |
| 0.008 | 3.0±0.3 | 2.4±0.4 | 2.3±0.5 | **1.7±0.2** | **1.8±0.4** | 2.3±0.7 | 2.5±0.4 | 3.9±0.8 | 4.5±0.9 | 5.5±1.0 |
| 0.010 | 2.8±0.4 | **1.8±0.5** | **1.9±0.4** | 2.1±0.2 | 2.3±0.6 | 3.6±0.8 | 4.1±0.6 | 4.9±0.9 | 7.5±0.7 | 6.3±0.8 |
| 0.012 | 2.5±0.1 | **1.4±0.2** | **1.6±0.4** | **1.8±0.2** | 3.1±0.7 | 4.4±1.1 | 5.4±0.5 | 5.6±1.4 | 6.6±0.8 | 7.5±0.7 |
| 0.014 | 2.6±0.3 | **1.5±0.3** | 2.5±0.5 | 2.2±0.5 | 3.9±0.6 | 5.0±0.8 | 7.1±1.2 | 6.3±0.6 | 8.3±0.7 | 8.3±0.5 |
| 0.016 | 2.6±0.4 | **1.3±0.4** | 2.0±0.3 | 3.4±1.3 | 4.4±0.9 | 6.1±1.0 | 7.0±1.1 | 6.9±0.8 | 8.0±0.6 | 8.9±0.8 |
| 0.018 | 2.3±0.2 | **1.3±0.4** | 2.8±0.8 | 4.1±0.6 | 5.1±0.9 | 6.9±1.0 | 7.1±1.3 | 8.3±0.5 | 9.1±0.9 | 9.3±0.4 |
| 0.020 | 2.0±0.6 | 2.0±0.5 | 3.2±1.2 | 4.4±0.8 | 6.3±1.3 | 7.2±1.1 | 7.4±0.6 | 9.1±0.8 | 10.2±0.9 | 10.0±0.6 |
| 0.022 | **1.7±0.3** | 2.0±1.0 | 3.9±1.3 | 5.3±1.0 | 6.8±1.0 | 7.5±1.1 | 8.8±0.6 | 9.1±0.8 | 10.7±0.5 | 10.0±0.3 |
| 0.024 | **1.9±0.2** | 3.3±0.9 | 4.3±1.1 | 6.0±0.9 | 7.0±0.3 | 8.5±0.7 | 9.3±0.4 | 10.1±0.7 | 10.7±0.5 | 10.8±0.6 |
| 0.026 | **1.5±0.4** | 2.7±1.2 | 4.3±0.6 | 7.0±0.7 | 8.3±0.4 | 9.0±0.4 | 9.8±0.7 | 10.1±0.7 | 10.7±0.6 | 11.6±0.4 |
| 0.028 | **1.3±0.2** | 3.8±0.6 | 4.9±0.7 | 7.2±0.7 | 8.8±1.0 | 8.8±0.9 | 9.5±0.5 | 10.7±0.4 | 10.9±0.2 | 11.3±0.4 |
| 0.030 | **1.3±0.4** | 4.0±0.4 | 5.0±0.4 | 7.1±0.9 | 8.2±1.0 | 9.2±0.6 | 10.3±0.5 | 10.6±1.0 | 10.9±0.4 | 11.8±0.4 |
| 0.032 | **1.5±0.5** | 6.1±1.2 | 6.2±1.1 | 7.3±0.6 | 9.1±0.8 | 9.9±0.5 | 10.6±0.4 | 10.5±0.8 | 11.2±0.6 | 12.0±0.3 |
| 0.034 | **1.9±0.6** | 5.1±0.8 | 5.9±0.7 | 8.2±0.7 | 9.0±0.3 | 10.4±0.4 | 10.6±0.6 | 11.2±0.4 | 11.9±0.5 | 12.1±0.4 |
| 0.036 | **1.5±0.2** | 6.4±0.5 | 6.8±0.6 | 7.9±1.0 | 9.3±1.0 | 10.4±0.4 | 10.9±0.3 | 11.3±0.5 | 11.9±0.3 | 12.0±0.3 |
| 0.038 | 2.2±1.6 | 5.1±0.8 | 6.1±0.6 | 8.2±1.0 | 9.6±0.8 | 10.8±0.5 | 10.9±0.5 | 11.6±0.3 | 11.8±0.4 | 12.3±0.5 |
| 0.040 | 2.2±1.1 | 7.3±1.6 | 7.6±0.4 | 8.7±0.4 | 9.3±0.7 | 10.4±0.5 | 11.4±0.4 | 11.7±0.6 | 12.3±0.2 | 12.2±0.3 |
| 0.042 | 2.2±0.8 | 6.4±1.2 | 7.6±0.8 | 9.3±0.8 | 10.0±0.8 | 10.4±0.5 | 11.6±0.3 | 12.3±0.4 | 12.0±0.5 | 12.5±0.5 |
| 0.044 | **1.8±0.6** | 6.8±1.0 | 6.9±0.9 | 9.6±0.5 | 10.3±0.4 | 11.0±1.0 | 11.1±0.3 | 12.0±0.3 | 12.4±0.4 | 12.6±0.3 |
| 0.046 | 2.1±0.3 | 7.4±0.7 | 8.1±0.8 | 9.4±0.3 | 10.4±0.9 | 11.3±0.4 | 11.6±0.5 | 12.1±0.4 | 12.4±0.2 | 12.7±0.2 |
| 0.048 | 3.2±1.1 | 6.6±1.0 | 7.5±1.3 | 9.9±0.4 | 10.1±0.4 | 11.3±0.2 | 12.2±0.4 | 12.0±0.5 | 12.3±0.3 | 12.5±0.3 |
| 0.050 | 3.5±1.0 | 7.7±0.8 | 8.6±0.3 | 10.0±0.3 | 10.6±0.6 | 11.1±0.5 | 12.4±0.2 | 12.5±0.4 | 12.6±0.3 | 12.8±0.2 |
| 0.052 | 3.6±0.7 | 8.8±0.9 | 8.1±0.6 | 10.0±0.8 | 11.0±0.4 | 11.6±0.7 | 12.7±0.2 | 12.5±0.1 | 12.8±0.4 | 13.3±0.1 |
| 0.054 | 3.8±1.5 | 7.3±0.9 | 8.5±0.3 | 10.1±0.9 | 11.3±0.3 | 11.7±0.2 | 12.7±0.3 | 12.9±0.2 | 12.8±0.2 | 13.0±0.3 |
| 0.056 | 3.8±1.3 | 8.8±1.0 | 9.1±0.8 | 10.5±0.7 | 11.1±0.7 | 11.8±0.4 | 12.4±0.2 | 12.6±0.4 | 13.1±0.1 | 13.0±0.2 |
| 0.058 | 4.1±1.0 | 9.1±1.1 | 9.3±1.3 | 10.7±0.5 | 10.9±0.2 | 11.9±0.3 | 12.1±0.4 | 12.7±0.4 | 13.1±0.2 | 13.2±0.3 |
| 0.060 | 4.1±1.0 | 8.8±0.4 | 9.1±0.9 | 10.8±0.5 | 11.2±0.5 | 11.8±0.3 | 12.5±0.2 | 12.8±0.2 | 13.0±0.3 | 13.2±0.2 |
| 0.062 | 4.1±1.7 | 9.2±1.0 | 9.1±0.6 | 10.9±0.5 | 11.9±0.5 | 12.1±0.3 | 12.4±0.2 | 12.9±0.4 | 13.0±0.3 | 13.3±0.1 |
| 0.064 | 4.5±1.2 | 8.6±0.5 | 9.7±0.4 | 11.1±0.8 | 11.7±0.2 | 12.3±0.5 | 12.6±0.3 | 13.1±0.2 | 13.3±0.2 | 13.4±0.2 |
| 0.066 | 4.7±0.3 | 9.5±0.9 | 9.2±0.4 | 11.0±0.5 | 12.0±0.3 | 12.1±0.3 | 12.8±0.4 | 12.9±0.2 | 13.3±0.2 | 13.3±0.2 |
| 0.068 | 4.7±0.7 | 9.2±1.0 | 9.7±0.7 | 11.0±0.7 | 11.7±0.4 | 12.8±0.4 | 12.5±0.6 | 13.0±0.2 | 13.4±0.1 | 13.4±0.2 |
| 0.070 | 5.4±1.5 | 9.5±0.9 | 10.1±0.6 | 10.8±0.4 | 12.3±0.3 | 12.4±0.4 | 12.5±0.7 | 13.1±0.4 | 13.3±0.2 | 13.5±0.1 |
| 0.072 | 5.3±1.2 | 9.1±0.8 | 10.1±0.4 | 11.7±0.5 | 12.2±0.6 | 12.5±0.4 | 13.0±0.4 | 13.3±0.2 | 13.2±0.2 | 13.7±0.2 |
| 0.074 | 5.8±1.0 | 10.1±0.8 | 10.3±0.4 | 11.4±0.9 | 12.0±0.4 | 12.5±0.4 | 12.7±0.3 | 13.0±0.4 | 13.3±0.3 | 13.7±0.1 |
| 0.076 | 5.2±1.3 | 9.8±0.5 | 10.6±0.5 | 11.3±0.9 | 12.3±0.3 | 12.7±0.5 | 13.0±0.2 | 13.2±0.2 | 13.5±0.3 | 13.7±0.2 |
| 0.078 | 5.9±0.5 | 10.3±0.5 | 9.8±0.6 | 11.8±0.1 | 12.1±0.1 | 12.7±0.4 | 13.2±0.3 | 13.3±0.3 | 13.6±0.1 | 13.7±0.2 |
| 0.080 | 5.6±0.7 | 10.1±0.1 | 10.6±0.4 | 11.4±0.4 | 12.3±0.5 | 13.0±0.3 | 13.1±0.1 | 13.4±0.3 | 13.5±0.4 | 13.7±0.2 |
| 0.082 | 4.3±1.3 | 10.3±0.8 | 10.2±0.7 | 11.8±0.5 | 12.5±0.4 | 12.8±0.3 | 13.1±0.3 | 13.4±0.2 | 13.5±0.3 | 13.8±0.2 |
| 0.084 | 6.7±0.9 | 10.6±0.3 | 10.9±0.2 | 11.8±0.3 | 12.5±0.3 | 13.0±0.3 | 13.3±0.2 | 13.4±0.3 | 13.6±0.2 | 13.8±0.2 |
| 0.086 | 4.9±0.6 | 10.2±0.6 | 11.0±0.4 | 11.9±0.3 | 12.4±0.2 | 12.6±0.5 | 13.0±0.3 | 13.5±0.2 | 13.7±0.2 | 13.9±0.2 |
| 0.088 | 5.8±1.0 | 10.7±0.6 | 10.9±0.2 | 11.7±0.6 | 12.5±0.2 | 13.0±0.5 | 13.3±0.2 | 13.6±0.3 | 13.6±0.1 | 13.9±0.1 |
| 0.090 | 6.6±1.4 | 10.2±0.6 | 11.1±0.4 | 12.1±0.3 | 12.6±0.5 | 13.0±0.2 | 13.5±0.1 | 13.4±0.2 | 13.6±0.2 | 13.8±0.2 |
| 0.092 | 7.0±1.0 | 10.4±0.6 | 11.1±0.7 | 12.2±0.3 | 12.8±0.3 | 13.0±0.2 | 13.3±0.2 | 13.5±0.3 | 13.7±0.2 | 13.8±0.2 |
| 0.094 | 7.9±0.5 | 10.2±0.2 | 11.2±0.6 | 12.3±0.2 | 12.5±0.3 | 12.8±0.3 | 13.2±0.2 | 13.5±0.2 | 13.6±0.2 | 13.9±0.2 |
| 0.096 | 6.7±0.5 | 10.9±0.6 | 11.1±0.2 | 12.2±0.5 | 12.8±0.2 | 13.1±0.3 | 13.4±0.3 | 13.4±0.2 | 13.8±0.3 | 13.9±0.1 |
| 0.098 | 7.6±0.5 | 10.7±0.5 | 11.1±0.2 | 12.2±0.3 | 12.6±0.3 | 13.0±0.4 | 13.3±0.2 | 13.6±0.2 | 13.7±0.2 | 14.0±0.1 |
| 0.100 | 8.2±1.0 | 10.8±0.3 | 11.3±0.8 | 11.9±0.4 | 13.0±0.2 | 13.3±0.3 | 13.4±0.1 | 13.6±0.2 | 13.8±0.1 | 13.8±0.2 |

# E  DISCUSSION

## E.1  BEYOND CONTINUITY

### E.1.1  FOR DISCONTINUOUS OBJECTIVE FUNCTIONS

We design experiments on discontinuous objective functions by adding random perturbation to synthetic functions in appendix B. For example, the perturbation is set to be

$$P(x) = \sum_{i=1}^{m} \epsilon_i \cdot \delta_{\mathcal{B}(x_i, 0.5)}(x).$$

$\mathcal{B}(x_i, 0.5)$ is the open ball centering at $x_i$ with radius equals to 0.5, with $x_i$, $i = 1, \dots, m$, randomly generated within the solution region. $\delta_{\mathcal{B}(x_i, 0.5)}(x)$ is an indicator function, equals to 1 when $x \in \mathcal{B}(x_i, 0.5)$ otherwise 0. The perturbations $\epsilon_i$ are uniformly sampled from $[0, 1]$ for every single ball center $x_i$, $i = 1, \dots, m$. The objective function is set to be

$$\tilde{f}(x) := f(x) + P(x),$$

which is lower semi-continuous. We use the same settings as in section 4.1 with dimension $n = 50$, the perturbation size $m = 5n$ and budget $T = 100n$. Similarly, region shrinking rate is set to be $\gamma = 0.9$ and region shrinking frequency is $\rho = 0.01$. Each of the algorithm is repeated 30 runs and the convergence trajectories of mean of the best-so-far value are presented in Figure 8. The numbers attached to the algorithm names in the legend of figures are the mean value of obtained minimum. It can be observed that the acceleration of "RACE-CARS" to "SRACOS" is still valid. Comparing with baselines, "RACE-CARS" performs the best on both convergence, and obtain the best optimal value. As we anticipated, the performance of "SRACOS" and "RACE-CARS" are almost impervious to discontinuity, whereas the other three baselines, whose convergence relies on the continuity, suffers from oscillation or early-stopping to different extent.

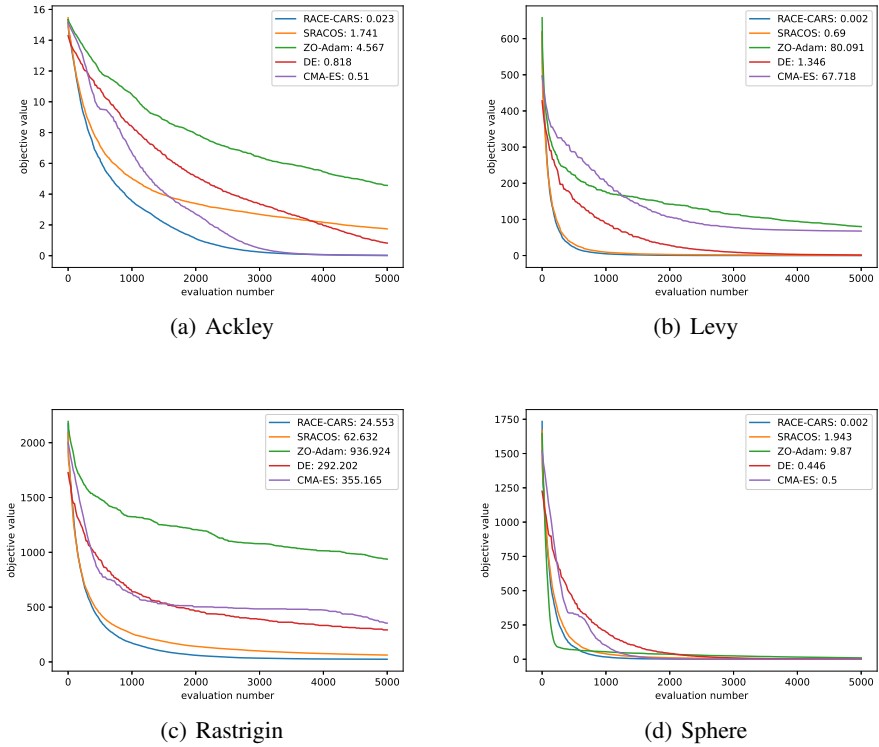

(a) Ackley

(b) Levy

(c) Rastrigin

(d) Sphere

Figure 8: Comparison on discontinuous objectives

### E.1.2 For discrete optimization

In order to switch "RACE-CARS" to discrete optimization, $Training$, $Region\text{-}shrinking$ and $Projection$ sub-procedures in Algorithm 1 should be modified. In all cases, we employ the discrete version of "RACOS" (Yu et al., 2016) for $Training$. Furthermore, we presume counting measure $\#$ as the inducing measure of probability space $(\Omega, \mathcal{F}, \mathbb{P})$, where $\mathbb{P}(B) := \#(B)/\#(\Omega)$ for all $B \in \mathcal{F}$. The $Region\text{-}shrinking$ and $Projection$ is therefore similar only to set the operator $\|\cdot\|$ return the count of each dimension of the region.

We design experiments on the following formulation:

$$\min \ f(x, y) \tag{4}$$
$$s.t. \ x \in \Omega_c$$
$$y \in \Omega_d,$$

where $\Omega_c$ is the continuous solution subspace and $\Omega_d$ is discrete. Equation 4 encompasses a wide range of continuous, discrete and mixed-integer programming problems. In our experiments, we specify equation 4 as a mixed-integer programming problem:

$$\min \ Ackley(x) + L^T abs(y)$$
$$s.t. \ x \in [-1, 1]^{n_1}$$
$$y \in \{-10, -9, \dots, 9, 10\}^{n_2},$$

where $L \in \mathbb{R}^{n_2}$ is sampled uniformly from $[1, 2]^{n_2}$, thus the global optimal value is 0. We choose the dimension of solution space as $n_1 = n_2 = 50$ and 250, the budget of function evaluation is set to be 3000 and 10000 respectively. Region shrinking rate is set to be $\gamma = 0.95$ and region shrinking frequency is $\rho = 0.01, 0.005$ respectively. Each of the algorithm is repeated 30 runs and the convergence trajectories of mean of the best-so-far value are presented in Figure 9. As results show, "RACE-CARS" maintains acceleration to "SRACOS" in discrete situation.

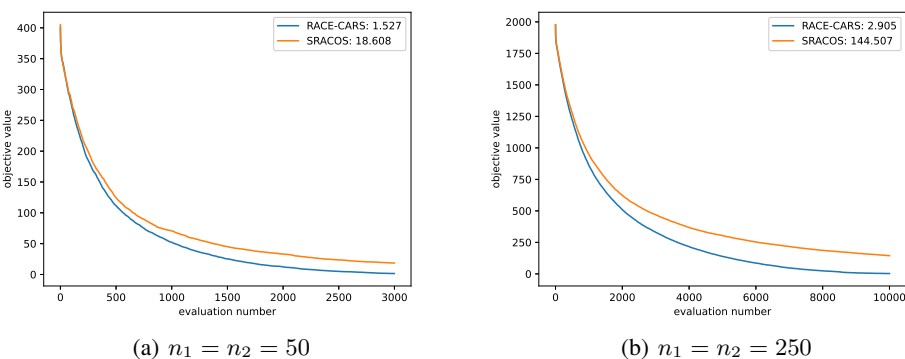

(a) $n_1 = n_2 = 50$          (b) $n_1 = n_2 = 250$

**Figure 9: Mixed-integer programming.**

### E.2 Ablation experiments

**(i) Relationship between shrinking frequency $\rho$ and dimension n.**

For Ackley on $\Omega = [-10, 10]^n$, we fix shrinking rate $\rho = 0.95$ and compare the performance of "RACE-CARS" between different shrinking frequency $\rho$ and dimension $n$. The shrinking frequencies $\rho$ ranges from 0.002 to 0.2 and dimension $n$ ranges from 50 to 500. The function calls budget is set to be $T = 30n$ for fair. Experiments are repeated 5 times for each hyperparameter and results are recorded in appendix D Table 1 and the normalized results is presented in heatmap format in Figure 10. The black curve represents the trajectory of best shrinking frequency with respect to dimension. Results in Figure 10 indicate the best $\rho$ is in reverse proportion to $n$, therefore maintaining $n\rho$ as constant is preferred.

**(ii) Relationship between shrinking factor $\gamma^{n\rho}$ and dimension n of solution space.**

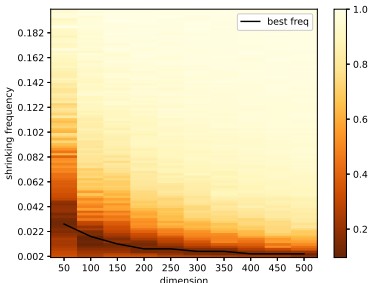

**Figure 10: Comparison of shrinking frequencies $\rho$ for Ackley on $\Omega = [-10, 10]^n$ with shrinking rate $\gamma = 0.95$.** The horizontal axis is dimension and the vertical axis is shrinking frequency. The heat of each pixel represents the y-wise normalized mean function value at the $30n$ step. The black curve is the best frequency in each dimension.

For Ackley on $\Omega = [-r, r]^n$, we compare the performance of "RACE-CARS" between different shrinking factors and radius $r$. Different shrinking factors are generated by varying shrinking rate $\gamma$ and dimension times shrinking frequency $n\rho$. We design experiments on 4 different dimensions $n$ with 4 radii $r$. The function calls budget is set to be $T = 30n$. Experiments are repeated 5 times for each hyperparameter and results are presented in heatmap format in Figure 11. According to the results, the best shrinking factor is insensitive to the variation of dimension. Considering that the best $n\rho$ maintains constant as $n$ varying, slightly variation of the corresponding best $\gamma$ is preferred. This observation is in line with what we anticipated as in section 4.

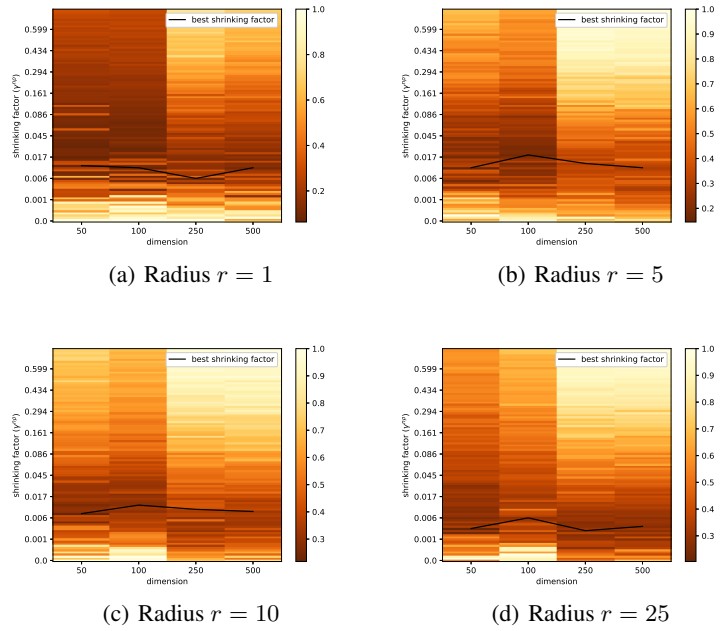

**Figure 11: Comparison of shrinking factor $\gamma^{n\rho}$ and dimension n of the solution space $\Omega = [-\mathbf{r}, \mathbf{r}]^\mathbf{n}$.** Results of different solution space radius are presented in each subfigure respectively. In each subfigure, the horizontal axis is the dimension and the vertical axis is shrinking factor. Each pixel represents the heat of y-wise normalized mean function value at the $30n$ step. The black curve is the best shrinking factor in each dimension.

**(iii) Relationship between shrinking factor $\gamma^{\mathbf{n}\rho}$ and radius r of solution space.**

For Ackley on $\Omega = [-r, r]^n$, we compare the performance of "RACE-CARS" between different shrinking factors and radius $r$. Different shrinking factors are generated by varying shrinking rate $\gamma$ and dimension times shrinking frequency $n\rho$. We design experiments on 4 different radii $r$ with 4 dimensions $n$. The function calls budget is set to be $T = 30n$. Experiments are repeated 5 times for each hyperparameter and results are presented in heatmap format in Figure 12. According to the results, the best shrinking factor $\gamma^{n\rho}$ should be decreased as radius $r$ increases.

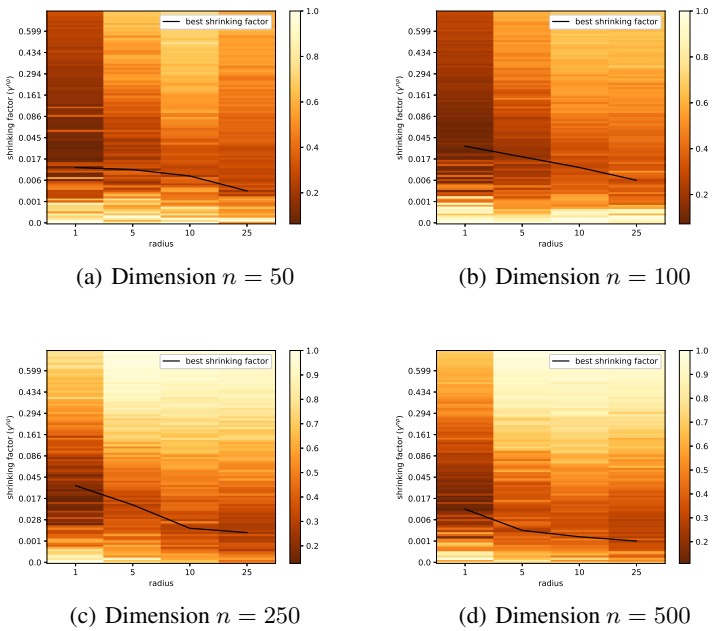

(a) Dimension $n = 50$      (b) Dimension $n = 100$

(c) Dimension $n = 250$      (d) Dimension $n = 500$

**Figure 12: Comparison of shrinking factor $\gamma^{n\rho}$ and radius r of the solution space $\Omega = [-r, r]^n$.** Results of different dimension are presented in each subfigure respectively. In each subfigure, the horizontal axis is the radius of solution space and the vertical axis is shrinking factor. Each pixel represents the heat of y-wise normalized mean function value at the $30n$ step. The black curve is the best shrinking factor of each solution space radius.

## F  PROOFS

**Theorem F.1.** Let $\mathbf{X}_t = \mathbf{X}_{h_t}$, assume that for $\epsilon > 0$, $\Omega_\epsilon$ is $\eta$-shattered by $h_t$ for all $t = r+1 \ldots, T$ and $\max\limits_{t=r+1,\ldots,T} \mathbb{P}(\{x \in \Omega \colon h_t(x) = 1\}) \leq p \leq 1$. Then for $0 < \delta < 1$, the $(\epsilon, \delta)$-query complexity is upper bounded by

$$\mathcal{O}\bigg( \max\{(\lambda\frac{\eta}{p} + (1-\lambda))^{-1}\big(\frac{1}{|\Omega_\epsilon|} \ln\frac{1}{\delta} - r\big) + r, T\}\bigg).$$

*Proof.* Let $\tilde{x} := \arg\min\limits_{t=1,\ldots,T} f(x_t)$, then

$$\Pr\left(f(\tilde{x}) - f^* > \epsilon\right)$$
$$= \mathbb{E}\left[\mathbb{I}_{\{\mathbf{Y}_1,\ldots,\mathbf{Y}_{T-1} \in \Omega_\epsilon^c\}} \mathbb{E}[\mathbb{I}_{\{\mathbf{Y}_T \in \Omega_\epsilon^c\}}|\mathcal{F}_{T-1}]\right]$$
$$= \mathbb{E}\left[\mathbb{I}_{\{\mathbf{Y}_1,\ldots,\mathbf{Y}_{T-2} \in \Omega_\epsilon^c\}} \mathbb{E}\left[\mathbb{I}_{\{\mathbf{Y}_{T-1} \in \Omega_\epsilon^c\}} \mathbb{E}[\mathbb{I}_{\{\mathbf{Y}_T \in \Omega_\epsilon^c\}}|\mathcal{F}_{T-1}]|\mathcal{F}_{T-2}]\right]\right]$$
$$= \cdots\cdots$$
$$= \mathbb{E}\left[\mathbb{I}_{\{\mathbf{Y}_1,\ldots,\mathbf{Y}_r \in \Omega_\epsilon^c\}} \mathbb{E}\left[\mathbb{I}_{\{\mathbf{Y}_{r+1} \in \Omega_\epsilon^c\}} \cdots \mathbb{E}\left[\mathbb{I}_{\{\mathbf{Y}_{T-1} \in \Omega_\epsilon^c\}} \mathbb{E}[\mathbb{I}_{\{\mathbf{Y}_T \in \Omega_\epsilon^c\}}|\mathcal{F}_{T-1}]|\mathcal{F}_{T-2}] \cdots |\mathcal{F}_r]\right]\right].$$

Where $\mathbb{I}_B(x)$ is the identical function on $B \in \mathcal{F}$ such that $\mathbb{I}_B(x) \equiv 1$ for all $x \in B$ and $\mathbb{I}_B(x) \equiv 0$ otherwise. At step $t \geq r+1$, since $\mathbf{X}_\Omega$ is independent to $\mathcal{F}_{t-1}$, it holds

$$\mathbb{E}[\mathbb{I}_{\{\mathbf{Y}_t \in \Omega_\epsilon^c\}}|\mathcal{F}_{t-1}] = \mathbb{E}[\mathbb{I}_{\{\lambda\mathbf{X}_t + (1-\lambda)\mathbf{X}_\Omega \in \Omega_\epsilon^c\}}|\mathcal{F}_{t-1}]$$
$$= \lambda(1 - \mathbb{E}[\mathbb{I}_{\{\mathbf{X}_{h_t} \in \Omega_\epsilon\}}|\mathcal{F}_{t-1}]) + (1-\lambda)(1 - |\Omega_\epsilon|).$$

Under the assumption that $\Omega_\epsilon$ is $\eta$-shattered by $h_t$, it holds the relation that

$$\mathbb{E}[\mathbb{I}_{\{\mathbf{X}_{h_t} \in \Omega_\epsilon\}}|\mathcal{F}_{t-1}] = \frac{\mathbb{P}(\{x \in \Omega_\epsilon \colon h_t(x) = 1\})}{\mathbb{P}(\{x \in \Omega \colon h_t(x) = 1\})} \geq \frac{\eta}{p}|\Omega_\epsilon|.$$

Therefore,

$$\mathbb{E}[\mathbb{I}_{\{\mathbf{Y}_t \in \Omega_\epsilon^c\}}|\mathcal{F}_{t-1}] = \lambda(1 - \mathbb{E}[\mathbb{I}_{\{\mathbf{X}_{h_t} \in \Omega_\epsilon\}}|\mathcal{F}_{t-1}]) + (1-\lambda)(1 - |\Omega_\epsilon|)$$
$$\leq 1 - \big(\lambda\frac{\eta}{p} + (1-\lambda)|\big)\Omega_\epsilon|.$$

Apparently, the upper bound of $\mathbb{E}[\mathbb{I}_{\{\mathbf{Y}_t \in \Omega_\epsilon^c\}}|\mathcal{F}_{t-1}]$ satisfies $0 < 1 - \big(\lambda\frac{\eta}{p} + (1-\lambda)\big)|\Omega_\epsilon| < 1$, thus

$$\mathbb{E}\left[\mathbb{I}_{\{\mathbf{Y}_t \in \Omega_\epsilon^c\}} \mathbb{E}[\mathbb{I}_{\{\mathbf{Y}_{t+1} \in \Omega_\epsilon^c\}}|\mathcal{F}_t]|\mathcal{F}_{t-1}\right] \leq \big(1 - (\lambda\frac{\eta}{p} + (1-\lambda))|\Omega_\epsilon|\big)\mathbb{E}[\mathbb{I}_{\{Y_t \in \Omega_\epsilon^c\}}|\mathcal{F}_{t-1}]$$
$$\leq \big(1 - (\lambda\frac{\eta}{p} + (1-\lambda))|\Omega_\epsilon|\big)^2.$$

Moreover,

$$\Pr\left(f(\tilde{x}) - f^* > \epsilon\right)$$
$$= \mathbb{E}\left[\mathbb{I}_{\{\mathbf{Y}_1,\ldots,\mathbf{Y}_r \in \Omega_\epsilon^c\}} \mathbb{E}\left[\mathbb{I}_{\{\mathbf{Y}_{r+1} \in \Omega_\epsilon^c\}} \cdots \mathbb{E}\left[\mathbb{I}_{\{\mathbf{Y}_{T-1} \in \Omega_\epsilon^c\}} \mathbb{E}[\mathbb{I}_{\{\mathbf{Y}_T \in \Omega_\epsilon^c\}}|\mathcal{F}_{T-1}]|\mathcal{F}_{T-2}] \cdots |\mathcal{F}_r]\right]\right]$$
$$\leq \big(1 - (\lambda\frac{\eta}{p} + (1-\lambda))|\Omega_\epsilon|\big)^{T-r}\mathbb{E}[\mathbf{Y}_1,\ldots,\mathbf{Y}_r \in \Omega_\epsilon^c]$$
$$= \big(1 - (\lambda\frac{\eta}{p} + (1-\lambda))|\Omega_\epsilon|\big)^{T-r}(1 - |\Omega_\epsilon|)^r$$
$$\leq \exp\left\{-\left((T-r)(\lambda\frac{\eta}{p} + (1-\lambda)) + r\right)|\Omega_\epsilon|\right\}.$$

In order that $\Pr\left(f(\tilde{x}) - f^* > \epsilon\right) \leq \delta$, it suffices that

$$\exp\left\{-\left((T-r)(\lambda\frac{\eta}{p} + (1-\lambda)) + r\right)|\Omega_\epsilon|\right\} \leq \delta,$$

hence the $(\epsilon, \delta)$-query complexity is upper bounded by

$$\mathcal{O}\bigg( \max\{(\lambda\frac{\eta}{p} + (1-\lambda))^{-1}\big(\frac{1}{|\Omega_\epsilon|} \ln\frac{1}{\delta} - r\big) + r, T\}\bigg).$$

$\square$

**Theorem F.2.** Consider Algorithm 1. Assume that for $\epsilon > 0$, $\Omega_\epsilon$ is $\eta$-shattered by $\tilde{h}_t$ for all $t = r+1 \ldots, T$. Let the region shrinking rate $0 < \gamma < 1$ and region shrinking frequency $0 < \rho < 1$, then for $0 < \delta < 1$, the $(\epsilon, \delta)$-query complexity is upper bounded by

$$\mathcal{O}\left(\max\{\left(\frac{\gamma^{-\rho} + \gamma^{-(T-r)\rho}}{2}\lambda\eta + (1-\lambda)\right)^{-1}\left(\frac{1}{|\Omega_\epsilon|}\ln\frac{1}{\delta} - r\right) + r, T\}\right).$$

*Proof.* Let $\tilde{x} := \underset{t=1,\ldots,T}{\arg\min} \; f(x_t)$, then

$$\Pr\left(f(\tilde{x}) - f^* > \epsilon\right)$$
$$=\mathbb{E}\left[\mathbb{I}_{\{\mathbf{Y}_1,\ldots,\mathbf{Y}_{T-1}\in\Omega_\epsilon^c\}}\mathbb{E}[\mathbb{I}_{\{\mathbf{Y}_T\in\Omega_\epsilon^c\}}|\mathcal{F}_{T-1}]\right]$$
$$=\mathbb{E}\left[\mathbb{I}_{\{\mathbf{Y}_1,\ldots,Y_{T-2}\in\Omega_\epsilon^c\}}\mathbb{E}\left[\mathbb{I}_{\{\mathbf{Y}_{T-1}\in\Omega_\epsilon^c\}}\mathbb{E}[\mathbb{I}_{\{\mathbf{Y}_T\in\Omega_\epsilon^c\}}|\mathcal{F}_{T-1}]|\mathcal{F}_{T-2}\right]\right]$$
$$=\cdots\cdots$$
$$=\mathbb{E}\left[\mathbb{I}_{\{\mathbf{Y}_1,\ldots,\mathbf{Y}_r\in\Omega_\epsilon^c\}}\mathbb{E}\left[\mathbb{I}_{\{\mathbf{Y}_{r+1}\in\Omega_\epsilon^c\}}\cdots\mathbb{E}\left[\mathbb{I}_{\{\mathbf{Y}_{T-1}\in\Omega_\epsilon^c\}}\mathbb{E}[\mathbb{I}_{\{\mathbf{Y}_T\in\Omega_\epsilon^c\}}|\mathcal{F}_{T-1}]|\mathcal{F}_{T-2}\right]\cdots|\mathcal{F}_r\right]\right].$$

At step $t \geq r + 1$, since $\mathbf{X}_\Omega$ is independent to $\mathcal{F}_{t-1}$, it holds

$$\mathbb{E}[\mathbb{I}_{\{\mathbf{Y}_t\in\Omega_\epsilon^c\}}|\mathcal{F}_{t-1}] =\mathbb{E}[\mathbb{I}_{\{\lambda\mathbf{X}_t+(1-\lambda)\mathbf{X}_\Omega\in\Omega_\epsilon^c\}}|\mathcal{F}_{t-1}]$$
$$=\lambda(1 - \mathbb{E}[\mathbb{I}_{\{\mathbf{X}_t\in\Omega_\epsilon\}}|\mathcal{F}_{t-1}]) + (1-\lambda)(1 - |\Omega_\epsilon|).$$

The expectation of probability that $\tilde{h}_t$ hits positive region is upper bounded by

$$\mathbb{E}\left[\mathbb{P}(\{x \in \Omega\colon \tilde{h}_t(x) = 1\})|\mathcal{F}_{t-1}\right] \leq \gamma^{(t-r)\rho}\mathbb{P}[\Omega] = \gamma^{(t-r)\rho}.$$

Under the assumption that $\Omega_\epsilon$ is $\eta$-shattered by $\tilde{h}_t$, it holds the relation that

$$\mathbb{E}\left[\mathbb{I}_{\{\mathbf{X}_t\in\Omega_\epsilon\}}|\mathcal{F}_{t-1}\right] =\frac{\mathbb{P}\left(\{x \in \Omega_\epsilon\colon \tilde{h}_t(x) = 1\}\right)}{\mathbb{E}\left[\mathbb{P}(\{x \in \Omega\colon \tilde{h}_t(x) = 1\})|\mathcal{F}_{t-1}\right]}$$
$$\geq\gamma^{-(t-r)\rho}\eta|\Omega_\epsilon|.$$

Therefore,

$$\mathbb{E}[\mathbb{I}_{\{\mathbf{Y}_t\in\Omega_\epsilon^c\}}|\mathcal{F}_{t-1}] \leq 1 - \left(\lambda\gamma^{-(t-r)\rho}\eta + (1-\lambda)|\right)\Omega_\epsilon|.$$

Moreover,

$$\Pr\left(f(\tilde{x}) - f^* > \epsilon\right)$$
$$=\mathbb{E}\left[\mathbb{I}_{\{\mathbf{Y}_1,\ldots,\mathbf{Y}_r\in\Omega_\epsilon^c\}}\mathbb{E}\left[\mathbb{I}_{\{\mathbf{Y}_{r+1}\in\Omega_\epsilon^c\}}\cdots\mathbb{E}\left[\mathbb{I}_{\{\mathbf{Y}_{T-1}\in\Omega_\epsilon^c\}}\mathbb{E}[\mathbb{I}_{\{\mathbf{Y}_T\in\Omega_\epsilon^c\}}|\mathcal{F}_{T-1}]|\mathcal{F}_{T-2}\right]\cdots|\mathcal{F}_r\right]\right]$$
$$\leq \prod_{t=r+1}^{T}\left(1 - \left(\lambda\gamma^{-(t-r)\rho}\eta + (1-\lambda)|\right)\Omega_\epsilon|\right)(1 - |\Omega_\epsilon|)^r$$
$$\leq \exp\left\{-\left(\sum_{t=r+1}^{T}\lambda\gamma^{-(t-r)\rho}\eta + (T-r)((1-\lambda)) + r\right)|\Omega_\epsilon|\right\}$$
$$= \exp\left\{-\left((T-r)(\frac{\gamma^{-\rho} + \gamma^{-(T-r)\rho}}{2}\lambda\eta + (1-\lambda)) + r\right)|\Omega_\epsilon|\right\}.$$

In order that $\Pr\left(f(\tilde{x}) - f^* > \epsilon\right) \leq \delta$, it suffices that

$$\exp\left\{-\left((T-r)(\frac{\gamma^{-\rho} + \gamma^{-(T-r)\rho}}{2}\lambda\eta + (1-\lambda)) + r\right)|\Omega_\epsilon|\right\} \leq \delta,$$

hence the $(\epsilon, \delta)$-query complexity is upper bounded by

$$\mathcal{O}\left(\max\{\left(\frac{\gamma^{-\rho} + \gamma^{-(T-r)\rho}}{2}\lambda\eta + (1-\lambda)\right)^{-1}\left(\frac{1}{|\Omega_\epsilon|}\ln\frac{1}{\delta} - r\right) + r, T\}\right).$$

$\square$

