# OpenReview forum: "A Region-Shrinking-Based Acceleration for Classification-Based Derivative-Free Optimization"
_ICLR.cc/2024/Conference — Submitted to ICLR 2024_

### Official Review · Reviewer_3VLh · 2023-10-17

**Soundness:** 3 good
**Presentation:** 2 fair
**Contribution:** 3 good
**Rating:** 6
**Confidence:** 2

**Summary:**

This paper proposes a new algorithm for classification-based derivative free optimization (DFO). This type of algorithm learns a classifier on the domain that predicts optimality, and alternates between updating that classifier and updating the best solution found so far. The paper has two main contributions: 1) a novel learning theoretical analysis and 2) a modification of an existing algorithm with better learning bounds achieved by shrinking the region where the best solution may be found so far. Experiments on DFO benchmarks and LLMs show that the proposed algorithm RACE-CARS improves over the modified algorithm SRACOS and CMA-ES (which is computationally expensive). Ablations of the hyperparameters are also done.

**Strengths:**

# Originality
The paper uses the learning theoretic concept of shattering to derive bounds for classification-based DFO, which is novel as far as I know.

# Quality
Experiments are done on a fairly large-scale LLM problem in addition to standard DFO benchmarks. Their results confirm the theoretical results in the paper and RACE-CARS is able to significantly improve over the previous classification-based DFO algorithm SRACOS. I cannot speak to the quality of the theory as I am not familiar with the relevant literature.

# Significance
Zero-order optimization is an increasingly popular area of research, as it does not require gradient computation. It is particularly useful for hyperparameter optimization, a common task that practitioners need to perform.

**Weaknesses:**

# Quality
There was no clear discussion of the computational complexity of the algorithms, only an observation that CMA-ES is more expensive than SRACOS or RACE-CARS. In addition, the LLM experiment only includes part of the baselines.

# Clarity
For clarity and readability, I think that some of the experimental results' graphs should be moved to the main paper, as they would be of more interest to practitioners.

Minor: Line numbers are not visible in Algorithms 1 and 2, but they are referred to on page 6.

**Questions:**

1. Does the use of the shattering rate remove all issues of the previous bound discussed in Section 3.1? Is the bound now tight? I think it would improve the clarity to provide a more explicit explanation.
2. How expensive is RACE-CARS compared to SRACOS (and other algorithms)? Some discussion and empirical study of the computational complexity would be helpful?
3. Did you try a BO algorithm as a baseline in the experiments? In addition, why was DE and ZO-Adam not included in the LLM experiment?

---

> ### Author Response · Authors · 2023-11-14
>
> Thanks for your kind comments and suggestion. We have reorganized the manuscript and supplemented more experiments, the new manuscript will release soon.
>
> # Answer to Q1:
>
> Thanks for your suggestions. Proof framework utilizing shattering rate removes the issue that computational complexity overflows the upper-bound. However, we cannot assert that the new upper bound is tight, since black-box functions are quite intricate. Meanwhile, we didn't make proofs about tightness. Nevertheless, the new bound is tighter in the more desirable cases, such as the counterexample we proposed.
>
> # Answer to Q2:
>
> For computational complexity upper bounds on function evaluations, you can refer to Theorem 3.1 and 3.2, for SRACOS and RACE-CARS respectively. As for numerical computation, neither SRACOS nor RACE-CARS involves matrix computation, the training and replacing sub-procedures consist of assignment operations and uniform sampling. You can refer to (Liu et al., 2017; 2019b.) for more running time comparisons or try code package in supplementary material if you interest. The only different operation of RACE-CARS is region shrinking, which consist of $4n$ boolean and assignment operations, the additional computational complexity can even be neglected.
>
> # Answer to Q3:
>
> Generally, BO is a good choice for functions with less oscillation in low dimension, but it fails in handerds dimensions (Bickel  Levina, 2004; Hall et al., 2005; Fan  Fan, 2008). In addition, randomly embedding BO was selected as a baseline in (Qian et al., 2016a;b; Hu et al., 2017), for RACOS and SRACOS respectively, and has been proven worse than baseline in our experiments. These are the reasons why we didn't try BO. We mentioned at footnote 2 in page 8, that DE and ZO-Adam are poor performed and we excluded these baselines in LLM for concision and explicit comparisons.

---

> > ### Comment · Reviewer_3VLh · 2023-11-22
> > **Thanks to the authors for your response**
> >
> > After reading the other reviews and responses, I have decided to keep my score.

---

### Official Review · Reviewer_ogoo · 2023-10-26

**Soundness:** 2 fair
**Presentation:** 3 good
**Contribution:** 2 fair
**Rating:** 3
**Confidence:** 4

**Summary:**

In this paper, the authors revisit the upper bound of the classification-based derivative-free optimization algorithm "RACOS",  and provide a query complexity analysis under the Hypothesis-Target  $\eta$-Shattering assumption.   In addition, the authors propose a sequential “RACE-CARS” method by introducing an adaptive projection sub-procedure to the previous "SRACOS" algorithm.

**Strengths:**

1. The classification-based derivative-free optimization is an interesting alternative compared with Bayesian optimization, and zeroth-order optimization.

2. Empirical results on four synthetic functions and black-box tuning for language-model-as-a-service on the SST2 dataset show faster convergence compared with baselines.

**Weaknesses:**

1. $\textbf{Exponentially Growing Complexity Bound}$

  The query complexity in Theorem 3.1 and 3.2 exponentially grows w.r.t. the problem dimension $n$.
For example, consider $\Omega =[-1,1]^n$  and $f(\boldsymbol{x}) = ||   \boldsymbol{x} ||_1 $, where $||  \cdot ||_1$ denote the $l_1$-norm,  then $| \Omega  _ {\epsilon}  |= \mathbb{ P } ( \Omega  _ {\epsilon}  ) \le \epsilon^n$ .  As a result, the term  $\frac{1}{| \Omega  _ {\epsilon}  |}  $ in the complexity bound  equals to  $\frac{1}{| \Omega  _ {\epsilon}  |} =  (\frac{1}{\epsilon})^n $, which grows exponentially for $\epsilon < 1$.

2. $\textbf{Impractical Assumptions}$

 The Hypothesis-Target  $\eta$-Shattering assumption in Definition 3.1 together with the $\gamma$-Shrinking assumption in Definition 2.3 are too strong to be practical.  The Hypothesis-Target  $\eta$-Shattering assumption requires the classifier prediction to have at least a $\eta$ overlap with the true $\epsilon$-solution domain $\Omega  _ {\epsilon}  $.   In the above example,  the size of $\Omega  _ {\epsilon}  $ decay exponentially w.r.t. the dimension $n$ , which requires the classifier to be very accurate or to have high positive prediction percentage ($h(x) =1$).

In addition, $\gamma$-Shrinking assumption requires the region of the positive prediction to have a $\gamma$-shrinking for all $t$.  As a result, the region of the positive prediction exponentially decays w.r.t. $t$,  i.e., $\gamma^t$.  However, the  Hypothesis-Target  $\eta$-Shattering assumption requires to have at least a $\eta$ overlap with the true $\epsilon$-solution domain $\Omega  _ {\epsilon}  $,  which requires a very strong classifier for high-dimensional problems. However, how to achieve a  strong classifier with limited training samples for high-dimensional problems is challenging.

3. $\textbf{No Convergence Guarantee}$.

The complexity analysis can not guarantee convergence as $t$ tends to infinity.   In Theorem 3.3, the left-hand side of the inequality decays exponentially fast w.r.t $t$. However,  the RHS of the inequality can not guarantee an exponential decay.

4. In the black-box tuning task, only the SST2 dataset is employed. The evaluation on a single dataset is not convincing enough to support the claim.

5. The paper is not well-organized. The experimental results are placed in the Appendix instead of the main paper.

**Questions:**

Q1. Could the authors please clarify the concern in the weakness section above?

Q2.  Could the authors please compare other datasets employed in (Sun et al. 2022) besides SST2 for better evaluation?

Q3. How to sample $(x_t, y_t) \sim \boldsymbol{Y}_t$ in Algorithm 2, what is the computational complexity or running time in this step?

---

> ### Author Response · Authors · 2023-11-14
>
> Thanks for your kind comments and suggestion. We have reorganized the manuscript and supplemented more experiments, the new manuscript will release soon.
>
> # Answer to Q1:
>
> ## Exponentially Growing Complexity Bound
>
> In order to prove universal acceleration, upper bounds in Theorem 3.1 and Theorem 3.2 are established only on the non-zero-measure assumption for objective $f.$ Consequently, the established bounds are inevitable to compromise for certain extreme objective functions. This compromise is quite common in convergence rate analysis. For instance, when we are talking about first order optimization algorithms for L-smooth objectives, the best convergence rate for convex functions is sublinear while for strongly convex functions is linear. As far as the authors' knowledge, for black-box objectives without additional assumption, curse of dimensionality is unavoidable. Fortunately, as Corollary 1 and 2 in Yu et al. (2016) show, the complexity upper bound of RACOS is polynomial about dimensionality $n$ (which we mentioned in introduction) when the objective is locally Holder continuous (the objective is not even point-wise continuous according to the definition), let alone the acceleration to SRACOS. In addition, in terms of the 4 Holder continuous synthetic functions, empirical results in appendix show the convergence complexity increases linearly w.r.t. dimensionality.
>
> ## Impractical Assumptions
>
> + Firstly, it is true that $\gamma$-shrinking assumption is to some extent irrational. But it is used in the former studies and is no longer needed in this paper. Rethinks of its plausibility is one of the reasons why we propose to shrink the region proactively.
>
> + Secondly, hypothesis-target $\eta$-shattering assumption coincides with error-target $\theta$-dependence. It can be proven that $\eta>0$ as long as $\theta<1-\mathbb{P}(R_t)$, meaning that our new assumption is at least not stronger than former studies. Moreover, the mechanism of training sub-procedure, i.e., RACOS in former studies, has shown the ability to maintain error-target $\theta$-dependence for small $\theta,$ in other words $\eta$-overlap with $\Omega_\epsilon.$ In fact, the trained classifier at each iteration entails to encompass as much proportion of true solution as possible, i.e.,  being more precise rather than being strong or accurate. RACOS itself is a radical training method, therefore satisfying the assumption is not bothering.
>
> + Lastly, the proposed DFO algorithm requires application-dependent hyper-parameters tuning. Introduction of supplementary hyper-parameters, which is reasonable for black-box exploration given the "no free lunch" theorem, indicates a concrete direction towards applicable DFO algorithms. In the present study, we address hyper-parameters selection from a practical perspective (see Discussion 5.2). Consolidation of the proposed theoretical analysis is considered as an important direction for our future works.
>
> ## No Convergence Guarantee
>
> To be frank, Theorem 3.3 plays as a sufficient but not necessary condition for Theorem 3.2, introducing a theoretic concept shrinking factor $\gamma^{n\rho}$ for empirical hyper-parameters tuning, rather than a convergence conclusion of the proposed algorithm. It is not trivial to establish a sufficient and necessary condition of Theorem 3.2, given the complexity of stochastic process coupling $Training$ and $replacing$ sub-procedures. However, according to our empirical analysis, region shrinking does not jeopardize the convergence of algorithm as long as hyper-parameters are well selected (see Discussion 5.2). Moreover, even an extremely conservative selection of hyper-parameters, i.e., $\gamma^\rho$ is extremely close to 1, achieves acceleration to SRACOS.
>
> # Answer to Q2:
>
> Experiments on other datasets have been updated. We will release the new manuscript soon. Please wait patiently.
>
> # Answer to Q3:
>
> The training sub-procedure RACOS outputs hypotheses $h_t$ with cubic positive region, and after orthogonal projection the positive region of new hypothesis $\tilde{h}_t$ is also cubic. Therefore, sampling $(x_t, y_t)\sim \mathbf{Y}_t$ is coordinate-wise uniform sampling, making it negligible compared with function evaluation.

---

### Official Review · Reviewer_6mCc · 2023-11-01

**Soundness:** 2 fair
**Presentation:** 1 poor
**Contribution:** 2 fair
**Rating:** 1
**Confidence:** 3

**Summary:**

The paper proposes a classification-based derivative-free optimization algorithm. It introduces the concept of hypothesis-target shattering rate and analyzes the computational complexity upper bound for the class of algorithms. By utilizing a random region-shrinking step and the revised upper bound, a new algorithm is presented.

**Strengths:**

It is very hard to read the paper.

**Weaknesses:**

The main results (i.e., numerical experiments) of the paper are not included in the main text of the submission. Note that reviewers are not required to read the appendix. This constitutes the primary rationale for my vote to reject the paper.

Moreover, the reviewer encounters difficulty in comprehending the algorithm descriptions due to the absence of several essential sub-procedure details, rendering the paper challenging to read.

**Questions:**

The paper is notably challenging to comprehend.

---

> ### Author Response · Authors · 2023-11-16
>
> Thanks for your comments and suggestions.
>
> We sincerely sorry for confusing you. We have reorganized the manuscript, elaborations on sub-procedures and parts of the numerical results are moved to the body section. Please leave comments if there is any comprehending problems.

---

### Official Review · Reviewer_boX1 · 2023-11-01

**Soundness:** 3 good
**Presentation:** 4 excellent
**Contribution:** 3 good
**Rating:** 6
**Confidence:** 3

**Summary:**

Authors propose a novel algorithm RACE-CARS for derivate-free global optimization that adds a mechanism of region-shrinking and theoretically show how this allows them to obtain an upper bound that is tighter than bound obtain to SRACOS algorithm.

**Strengths:**

Well written with rigorous theoretical analysis with experimental evaluation comparing with other approaches.

New theoretical bounds showing superiority of the proposed algorithm compared to the baseline and new general bound on complexity under assumptions of $\eta$-shuttering.

**Weaknesses:**

As I understood (correct me if am wrong) SRACOS and it's bound derived without assumption of Holder continuity which is the case of Theorem 3.2 for algorithms with shrinking, while sufficient conditions for $\eta$-shuttering in Theorem 3.3. are established only for Holder continuous, which makes results a bit less general as of SRACOS. Evaluation is done only on functions that satisfy this (even if we don't know their constants) -- it is interesting to see also evaluation on $f$ that do not satisfy Holder-continuity.

**Questions:**

Since there is Holder-continuity assumed, then say SGLD with 0-th order gradient estimator should work (theoretically) [1] -- interesting to see what authors think about that.

[1] Niladri Chatterji, Jelena Diakonikolas, Michael I Jordan, and Peter Bartlett. Langevin monte carlo without smoothness.

---

> ### Author Response · Authors · 2023-11-14
>
> Thanks for your kind comments and suggestion.
>
> # Answer to Weakness:
>
> Just as you mentioned, the upper bound of SRACOS and RACE-CARS, i.e., Theorem 3.1 and Theorem 3.2 respectively, are established both under the assumption of $\eta$-shattering rather than dimensionally local Holder continuity. In our paper, Theorem 3.3 gives a lower bound of $\gamma^\rho$ to maintain the same shattering rate of ``RACE-CARS'' and SRACOS, playing as a theoretical guidance for hyper-parameters selection when the objective is dimensionally locally Holder continuous. However, the generality of RACE-CARS is not less than SRACOS, we list the reasons below:
> + We adopt dimensionally locally Holder continuous as assumption for mathematical considerations, with which $\eta$-shattering can be simplified analytically. RACE-CARS achieves acceleration to SRACOS as long as $\eta>0,$ rather than limited in Holder continuous situation.
> + From practical standpoint, neither of the upper bounds of SRACOS and RACE-CARS are polynomial about $t,$ whereas Yu et al. (2016) gave a proof saying that RACOS converges to an $\epsilon$-solution in polynomial time when the objective is locally Holder continuous. Therefore, assumption for sufficient conditions in Theorem 3.3 is in line with what SRACOS needs.
> + Assumption on dimensionally local Holder continuity is a weakened equivalence to condition $|\Omega_\epsilon|>0$ for all $\epsilon>0.$ The latter one is needed to avoid discontinuity at global minima, such as the case $f(x)=0$ when $x=0$ otherwise $f(x)=1,$  which is literally impossible to solve with DFO algorithms. Additionally, according to the Definition 3.2, dimensionally local Holder continuity implies the objective is upper and lower enveloped by certain continuous functions, while discontinuities can still happen within the region.
>
> # Answer to Question:
>
> As aforementioned, point-wise continuity is not necessary for RACE-CARS in theoretical viewpoint, whereas without which DFO methods with gradient estimator may fail. Nevertheless, we appreciate the inspiring suggestions, we have supplemented experiments on discontinuous objectives, the new manuscript will release soon.

---

### Author Response · Authors · 2023-11-16
**Manuscript Update**

Dear Reviewers:

Thanks for your kind comments and suggestions. We have updated our manuscript.

We make modifications in the following aspects:

+ For clarity and readability, we reorganize the paper. We put several figures in the body section, including illustrations for $projection$ sub-procedure and parts of the numerical results in section 4.1, 4.2. Additionally, the detailed discussions is moved to appendix E.
+ We supplement experiments on datasets Yelp polarity, AG's News and RTE for LLM problem.
+ We supplement experiments on discontinuous objective functions and mixed-integer programming. Details can be found in appendix E.1.
+ We elaborate some concerning issues by the reviewers in section 5.
+ We correct some typos and statements. **Please attention we correct the misused Figure 4(a).**

---

### Meta-Review · Area_Chair_rdXX · 2023-12-09

**Metareview:**

In this paper, the authors revisit the upper bound of the classification-based derivative-free optimization algorithm "RACOS", and provide a query complexity analysis under a newly introduced Hypothesis-Target $\eta$-Shattering assumption. In addition, the authors propose a sequential “RACE-CARS” method by introducing an adaptive projection sub-procedure to the previous "SRACOS" algorithm.

While the proposed approach seems promising, the paper currently lacks clarity in its current form, as pointed out by several reviewers. A reviewer also pointed out the impractical assumptions and lack of convergence guarantees.

The experimental results were originally in the appendix and have been moved to the main text during the rebuttal. We therefore believe that another round of reviews is necessary for this paper.

**Justification For Why Not Higher Score:**

The paper currently lacks clarity in its current form, as pointed out by several reviewers. A reviewer also pointed out the impractical assumptions and lack of convergence guarantees.

The experimental results were originally in the appendix and have been moved to the main text during the rebuttal. We therefore believe that another round of reviews is necessary for this paper.

**Justification For Why Not Lower Score:**

N/A

---

### Decision · Program_Chairs · 2024-01-16

Reject